# GP73 is a TBC-domain Rab GTPase-activating protein contributing to the pathogenesis of non-alcoholic fatty liver disease without obesity

Yumeng Peng[1,2,8], Qiang Zeng [3,8], Luming Wan [1,8], Enhao Ma[4,8], Huilong Li[1,2,8], Xiaopan Yang [1,8], Yanhong Zhang[1], Linfei Huang[1], Haotian Lin[1], Jiangyue Feng[1], Yixin Xu[1,2], Jingfei Li[1], Muyi Liu[1], Jing Liu[1], Changqin Lin[5], Zhiwei Sun[5], Gong Cheng[4], Xuemiao Zhang[1,6], Jialong Liu[1,6], Dongrui Li[1], Meng Wei[1,2], Yunhai Mo[1,2], Xuetao Mu[6], Xiaowei Deng[6], Dandan Zhang[6], Siqing Dong[6], Hanqing Huang[1], Yi Fang[7,9], Qi Gao[5,9], Xiaoli Yang[6,9], Feixiang Wu [2,9], Hui Zhong [1,9] & Congwen Wei [1,9 ✉]

The prevalence of non-obese nonalcoholic fatty liver disease (NAFLD) is increasing worldwide with unclear etiology and pathogenesis. Here, we show GP73, a Golgi protein upregulated in livers from patients with a variety of liver diseases, exhibits Rab GTPase-activating protein (GAP) activity regulating ApoB export. Upon regular-diet feeding, liver-GP73-high mice display non-obese NAFLD phenotype, characterized by reduced body weight, intrahepatic lipid accumulation, and gradual insulin resistance development, none of which can be recapitulated in liver-GAP inactive GP73-high mice. Common and specific gene expression signatures associated with GP73-induced non-obese NAFLD and high-fat diet (HFD)-induced obese NAFLD are revealed. Notably, metformin inactivates the GAP activity of GP73 and alleviates GP73-induced non-obese NAFLD. GP73 is pathologically elevated in NAFLD individuals without obesity, and GP73 blockade improves whole-body metabolism in non-obese NAFLD mouse model. These findings reveal a pathophysiological role of GP73 in triggering non-obese NAFLD and may offer an opportunity for clinical intervention.

[1] Beijing Institute of Biotechnology, Academy of Military Medical Sciences (AMMS), Beijing, China. [2] Department of Hepatobiliary Surgery, Affiliated Tumor Hospital of Guangxi Medical University, Nanning, China. [3] Health management Institute, The Second Medical Center and National Clinical Research Center for Geriatric Diseases, Chinese PLA General Hospital, Beijing, China. [4] Tsinghua-Peking Center for Life Sciences, School of Medicine, Tsinghua University, Beijing, China. [5] Beijing Sungen Biomedical Technology Co., Ltd., Beijing, China. [6] Department of Clinical Laboratory, the Third Medical Center, Chinese PLA General Hospital, Beijing, China. [7] Department of Endocrinology, the Fifth Medical Centre, Chinese PLA General Hospital, Beijing, China. [8] These authors contributed equally: Yumeng Peng, Qiang Zeng, Luming Wan, Enhao Ma, Huilong Li, Xiaopan Yang. [9] These authors jointly supervised this work: Yi Fang, Qi Gao, Xiaoli Yang, Feixiang Wu, Hui Zhong, Congwen Wei. ✉email: weicw@yahoo.com

Nonalcoholic fatty liver disease (NAFLD) is the most common chronic liver disease, with a prevalence of over 25% in the global population, and can progress from simple steatosis to nonalcoholic steatohepatitis (NASH), fibrosis, cirrhosis, hepatocellular carcinoma (HCC), and death[1,2]. Although NAFLD is closely associated with an increased body mass index (BMI) and diabetes, it does occur in lean individuals with normal BMI values, which is referred to as lean or non-obese NAFLD[3]. Non-obese NAFLD constitutes just over 40% of the NAFLD population. Individuals with non-obese NAFLD are metabolically unhealthy and have a higher risk of type 2 diabetes mellitus (T2DM), an accelerated development of advanced fibrosis and a greater incidence of severe liver disease than those with obese NAFLD. Liver-related mortality is almost two times higher in the non-obese NAFLD population than in the obese NAFLD population. Non-obese NAFLD occurs in children and adults of all ethnicities[4]. Therefore, risk factors other than obesity may play a key role in the pathophysiology of NAFLD in the non-obese population. Dietary intake, such as fructose-sweetened beverages, soft drinks, and increased dietary cholesterol, may play an important role in the pathogenesis of non-obese NAFLD[5,6]. Specific gut microbiota compositions have been observed in lean patients with NAFLD[7]. However, the characteristics and risk factors for non-obese NAFLD are not well documented. Thus, elucidating the specific mechanisms leading to the development and worse outcome of individuals with non-obese NAFLD will promote the ability of healthcare systems to develop appropriate guidance and interventions to treat this liver disease.

NAFLD comprises a spectrum of conditions associated with lipid accumulation in hepatocytes. Intestinal barrier damage, including the destruction of the intestinal epithelium, the growth of bacteria in the small intestine, and an increase in the level of lipopolysaccharide (LPS), is associated with hepatocyte lipid accumulation and apoptosis[8]. The abnormal accumulation of lipids and apolipoproteins can also result from changes in the production, conversion, catabolism, or export of lipoprotein particles[9]. Very-low-density lipoproteins (VLDLs) are large particles (~30–80 nm) that are primarily composed of triglycerides (TGs), phospholipids, and cholesterol (CHO). The lipid cores of these particles are decorated with apolipoprotein B (ApoB). These lipid particles are synthesized and assembled in hepatocytes and then secreted into the plasma to be delivered to other organs through the circulation[10,11]. VLDLs' assembly and secretion from the liver play important roles in controlling the plasma levels of TGs and CHO, and defects in the export of VLDLs impair lipid homeostasis and predispose hepatocytes to the overproduction of reactive oxygen species (ROS), endoplasmic reticulum (ER) stress, and the induction of autophagy[12]. Alterations in multiple pathways, including the insulin response, β-oxidation, lipid storage and transport, and autophagy pathways, are closely interrelated and are indistinguishable with regard to their contributions to NAFLD and HCC[13,14].

How these hepatic cargoes are secreted from living cells remains an open question. Rab GTPases form the largest family of small guanosine triphosphate (GTP)-binding proteins and regulate multiple steps in eukaryotic vesicle trafficking[15]. Several Rab GTPases have been identified as regulators of hepatic lipoprotein secretion[16,17]. In particular, inhibition of Rab23 and Rab1b differentially affects the secretion of ApoE, ApoB100 and albumin[18]. Rabs cycle between a guanosine diphosphate (GDP)-bound inactive state and a GTP-bound active state. The exchange of GDP for GTP in Rabs is catalyzed by guanine nucleotide exchange factor, whereas GTP hydrolysis is facilitated by GTPase-activating protein (GAP)[19]. Most known GAPs for Rab GTPases share a homologous catalytic Tre-2/Bub2/Cdc16 (TBC) domain employing a conserved arginine finger and a unique catalytic glutamine residue (the "glutamine finger") to catalyze GTP hydrolysis[20,21].

GP73 is a type II transmembrane protein with a single N-terminal transmembrane domain and an extensive C-terminal coiled-coil domain located on the luminal surface of the Golgi apparatus[22]. All resident Golgi proteins are peripheral or integral membrane proteins, and the majority are involved in the intracellular modification of secretory proteins[23]. GP73 has no homology to the known glycosyltransferases or to any of the known nucleotide, sugar, or ATP transporters of the Golgi apparatus. GP73 expression is quite low in normal liver tissue but increases in response to liver damage, viral infection, or ER stress[23–25]. The physiological functions of GP73 as a Golgi-resident protein and the biological consequences of chronic increases in hepatocyte GP73 levels remain largely elusive.

In this study, we systematically dissected how GP73 reshapes the lipid metabolism of hepatocytes, promotes the onset of non-obese NAFLD, and accelerates its progression. Mice with chronic GP73 upregulation in hepatocytes exhibited a metabolic phenotype that has almost all the hallmarks of NAFLD without obesity in human patients. Treatment with metformin to inhibit the GAP activity of GP73 efficiently blocked the non-obese NAFLD phenotype induced by GP73. By revealing common and specific features of GP73-induced non-obese NAFLD, our study provides a potent therapeutic target and a potential diagnostic marker for non-obese NAFLD.

## Results

**GP73 harbors potent TBC-domain GAP activity that preferentially targets Rab23.** Research has revealed that most known GAPs for Rab GTPases share a homologous catalytic TBC (Tre-2/Bub2/Cdc16) domain, which employs conserved arginine and glutamine fingers (Fig. 1a). Subsequent analysis of the amino acid sequences of GP73 revealed a conserved TBC GAP domain containing a unique glutamine-finger motif (VxQ) in addition to the catalytic arginine-finger motif (VxxDxxR). We hypothesized that GP73 may function as a GAP catalyzing GTP hydrolysis for Rab GTPases. To test this hypothesis, we investigated the ability of recombinant GP73 protein to accelerate GTP hydrolysis. As GP73 is a Golgi-resident protein and traffics to the cell surface[26], we selected Rab GTPases mainly located at the plasma and intracellular membranes[27]. Among the 13 assayed mammalian Rab GTPases, GP73 showed the highest GAP activity toward Rab23 (Fig. 1b). Several other Rabs, including Rab13 and Rab20, also served as substrates of GP73 with lower catalytic efficiency. The catalytic efficiency ($k_{cat}/K_M$) of GP73 toward Rab23 was approximately $2.51 \times 10^5 \text{ M}^{-1}\text{ s}^{-1}$ (Fig. 1c, d). Importantly, the R248K Q310A mutant of GP73 (GP73-RQ) was completely unable to catalyze GTP hydrolysis on Rab23 (Fig. 1e). Thus, GP73 is a bona fide GAP with active enzyme sites at R248 and Q310.

**GP73 reduces ApoB secretion in a GAP activity-dependent manner.** Given that Rab23 regulates cargo transport from the ER to the Golgi and the secretion of lipoprotein, we hypothesized that the inhibition of Rab23 GTPase activity by GP73 might impact hepatocyte lipoprotein secretion[18,28]. To test this hypothesis, we investigated the involvement of GP73 in the secretion of hepatic cargo. The secretion efficiency fraction was calculated as the ratio between the amount of cargo that was secreted and the total amount of cargo (Supplementary Fig. 1a, b). Indeed, WT GP73 impaired the secretion of ApoB and ApoB100, whereas GP73-RQ abolished these effects (Fig. 1f, g). Notably, GP73 promoted the secretion of ApoE and albumin, but did not affect the secretion of ApoA1 or ApoB48 (Supplementary Fig. 1c–f). GP73-RQ showed the same activity with regard to the

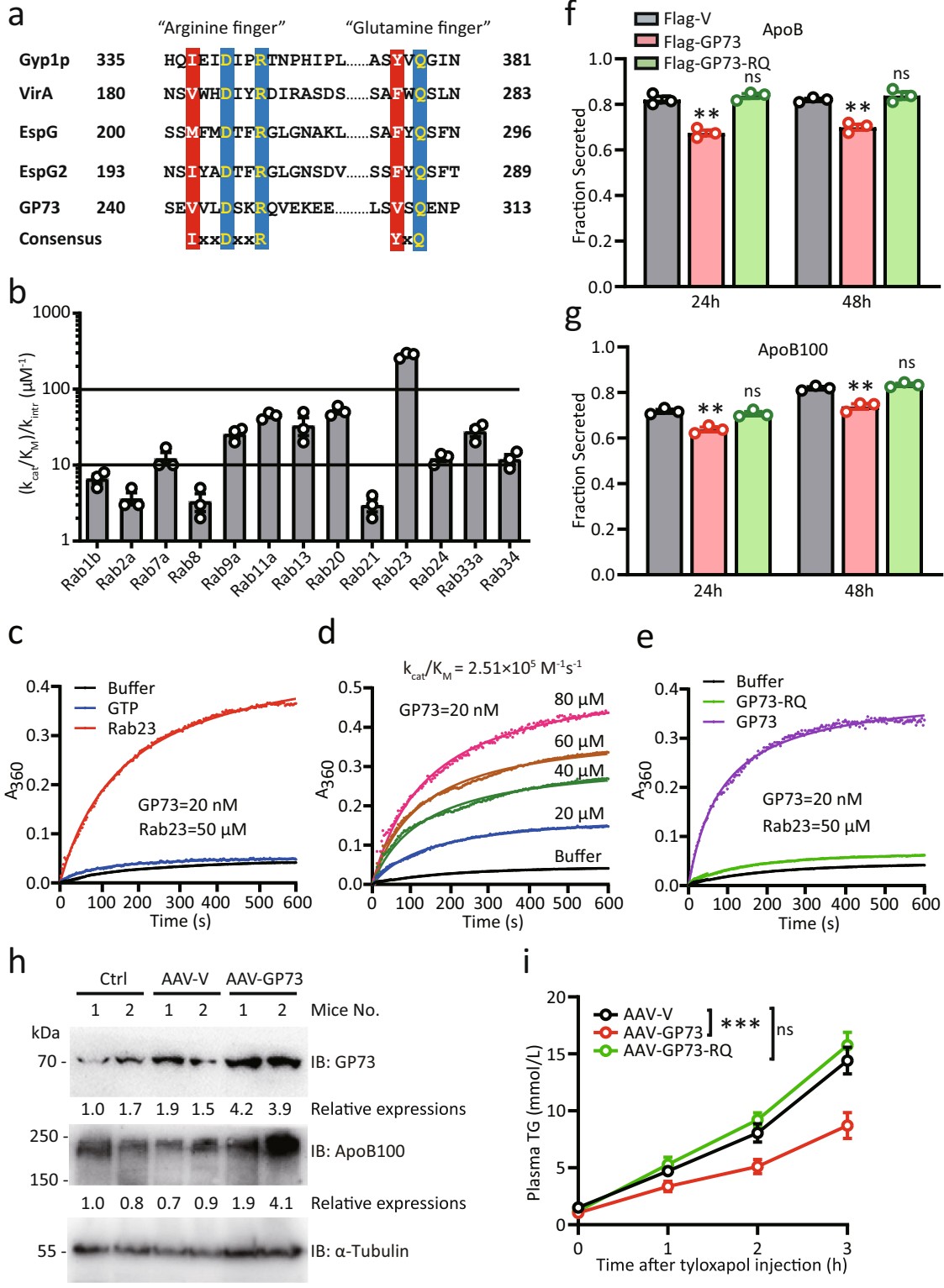

secretion of albumin, ApoE, ApoA1, and ApoB48 (Supplementary Fig. 1c–f). Together, these results indicate that the effects of GP73 on the export of the hepatic lipoproteins ApoB and ApoB100 are critically dependent on the GAP activity of GP73.

We then tested the regulation of cargo secretion by GP73 in vivo. To this end, a hepatocyte-specific adeno-associated virus (AAV) expressing an empty vector (AAV-V) or an AAV expressing GP73 (AAV-GP73) was injected into the tail veins of 8-week-old C57BL/6 mice. At 3 weeks after AAV

administration, AAV-GP73-induced GP73 expression occurred mainly in the liver, indicating that tail vein injection essentially limited the adenovirus target location to the liver (Supplementary Fig. 2a). In addition, the majority of GP73 expression was identified in hepatocytes from the livers of AAV-GP73-infected mice (Supplementary Fig. 2b). In this setting, significantly increased ApoB100 expression in livers from AAV-GP73 mice was revealed (Fig. 1h). By injecting tyloxapol, an inhibitor of lipoprotein lipase[29], we observed that AAV-GP73 mice had

**Fig. 1 GP73 harbors potent TBC-domain GAP activity that inhibits VLDLs' secretion. a** Comparison of amino acid sequences around R248 and Q310 in GP73 and equivalent residues in Gyp1p, VirA, EspG, and EspG2 with dual-finger catalytic motifs in the TBC domain. **b** GAP activity profiles of GP73 for a panel of 13 mammalian Rabs. The catalytic efficiency ($k_{cat}/K_M$) of GP73-catalyzed GTP hydrolysis relative to the intrinsic GTP hydrolysis rate constant was determined for each Rab. $n = 3$ independent biological experiments. Data are presented as mean ± SEM. **c, d** Kinetic analysis of GP73 GAP activity toward Rab23. The $k_{cat}/K_M$ determined by using a Lineweaver-Burk plot is listed above the activity curves. $n = 3$ independent biological experiments. **e** Effects of R248K and Q310A mutations in GP73 (GP73-RQ) on GP73 GAP activity. $n = 3$ independent biological experiments. **f, g** ApoB (**f**) and ApoB100 (**g**) secretion efficiency of Huh-7 cells transfected with Flag-vector (Flag-V), Flag-GP73, or Flag-GP73-RQ mutant at the indicated time points after transfection. Secretion efficiency was calculated as the fraction secreted, defined as the ratio between the amounts of cargo that was secreted and the total amount of cargo (secreted plus cell-associated cargo) present in a well. $n = 3$ independent biological experiments. Differences between the two groups were evaluated using two-tailed Student's $t$-test. Data are presented as mean ± SEM. ns, no statistical significance; $^{**}P < 0.01$. **h** GP73 and ApoB100 protein levels in mice livers at week 3 after the injection of AAV-V or AAV-GP73 ($n = 2$ per group). α-Tubulin was used as the equal loading control. Relative expression was calculated as the fold change in expression relative to the expression in No. 1 control mice. **i** TG concentrations at the indicated time points after blood sampling of AAV-V-, AAV-GP73-, or AAV-GP73-RQ-injected mice fasted for 4 h and then intravenously administered tyloxapol (400 mg/kg; $n = 6$ per group). Differences between the three groups were evaluated using two-way ANOVA and Bonferroni's post hoc analysis. Data are presented as mean ± SEM. ns, no statistical significance; $^{***}P < 0.001$.

markedly reduced plasma TG levels (indicative of newly secreted VLDLs particles) at 1, 2, and 3 h after tyloxapol injection, which corresponded to a 1-fold reduction in the VLDL-TG production rate in AAV-GP73-treated mice compared to AAV-V control mice (Fig. 1i). In contrast, hepatic expression of the AAV-GP73-RQ mutant did not affect VLDLs' secretion (Fig. 1i). Together, these data indicate that GP73 inhibits VLDLs' secretion in a manner dependent on its GAP activity.

**Chronic elevations in hepatocyte GP73 trigger non-obese NAFLD in mice.** Since impaired VLDLs' export is closely related to dyslipidemia, we then wanted to assess the metabolic consequences of chronic GP73 expression in hepatocytes. To this end, AAV-V and AAV-GP73 mice were fed a regular diet for 6 months, and liver tissues were then collected. Compared with the control mice, the liver-GP73-high animals developed slightly enlarged livers and exhibited increased liver-to-body weight and spleen-to-body weight ratio as well as massive lipid accumulation with indications of hepatocyte ballooning and a lower degree of fibrosis (Fig. 2a and Supplementary Fig. 3a–c). This finding coincided with the increased hepatic TGs and CHO levels in liver-GP73-high mice (Fig. 2b, c). Importantly, total TG and CHO levels in plasma from liver-GP73-high mice were dramatically reduced (Fig. 2d and Supplementary Fig. 3d), likely as a result of impaired VLDLs' secretion in the liver. The levels of circulating alanine transaminase (ALT) and aspartate transaminase (AST), which are measures of liver damage, were increased gradually and significantly at 6 and 12 months after AAV-GP73 injection (Fig. 2e and Supplementary Fig. 3e). Interestingly, the liver-GP73-high mice exhibited a reduced body weight and continued to lose weight throughout the experimental duration compared with the control mice (Fig. 2f and Supplementary Fig. 3f). From month 5 to 6 after AAV injection, the food intake in liver-GP73-high mice was similar to that in the control mice (Supplementary Fig. 3g).

We then attempted to determine whether lipotoxicity in the context of chronic GP73 upregulation relates to abnormalities in glucose metabolism. At 1.5 months after AAV injection, the fasting glucose levels were similar between liver-GP73-high mice and control mice (Supplementary Fig. 3h). However, liver-GP73-high mice exhibited higher fasting blood glucose levels than control mice at month 4 after AAV injection, and the difference persisted until month 6 before sacrifice (Fig. 2g, h). These liver-GP73-high animals began to develop severe glucose intolerance starting at 4.5 months after AAV-GP73 injection (Fig. 2i and Supplementary Fig. 3i, j). In particular, the plasma levels of cytokines, including IL-6, TGF-β, IL-1β, and IFN-γ, were significantly elevated in AAV-GP73-injected mice (Fig. 2j–m).

Therefore, we conclude that chronic increases in hepatocyte GP73 levels trigger non-obese NAFLD in mice.

**Induction of non-obese NAFLD by GP73 is highly dependent on its GAP activity.** We subsequently assessed whether non-obese NAFLD resulting from chronic increases in hepatocyte GP73 is primarily dependent on GP73 GAP activity. AAV-V, AAV-GP73, or AAV-GP73-RQ constructs were injected into the tail veins of 8-week-old C57BL/6 mice and fed a regular diet for 4.5 months. In contrast to the reduced plasma LDL, TG, and CHO levels and increased hepatic TG and CHO levels observed in liver-GP73-high mice, mice injected with GAP-inactive GP73 exhibited no changes in plasma or hepatic lipid levels (Fig. 3a–e). Notably, mice injected with the GP73 GAP mutant did not show any signs of liver damage (Fig. 3f, g).

To determine whether GP73 GAP activity regulates glucose hemostasis, we assessed fasting glucose levels over time after AAV injection. Remarkably, despite the substantial increases in fasting glucose levels observed in mice expressing WT GP73, mice expressing GP73-RQ were not hyperglycemic throughout the testing period (Fig. 3h–j). While AAV-GP73-injected mice exhibited impaired glucose tolerance, mice injected with GP73-RQ showed normal glucose uptake at 4.5 months after injection (Fig. 3k). Therefore, the GAP activity of GP73 is indispensable for its effect on metabolism.

**Gene expression signatures in non-obese NAFLD induced by GP73.** In searching for common and specific features of transcriptional changes associated with non-obese NAFLD induced by GP73 and obese NAFLD induced by high-fat diet (HFD), we analyzed gene expression in the livers isolated from mice 12 months after AAV-V or AAV-GP73 injection and in the livers isolated from AAV-V control mice fed a HFD for 12 months. Over 2000 differentially expressed genes (DEGs) in the livers of AAV-GP73-injected mice and over 1500 DEGs in the livers of HFD-fed mice were identified (Fig. 4a and Supplementary Fig. 4a). Gene Ontology (GO) enrichment analysis showed that the most significant biological processes affected by GP73 were metabolic process (Supplementary Fig. 4b). Further assays of metabolism process signatures revealed significant enrichment in fatty acid (FA) biosynthesis and oxidation process, tricarboxylic acid (TCA) cycle, protein metabolic process, lipid biosynthetic process, and triglyceride metabolic process in the livers from AAV-GP73 mice (Fig. 4b). In particular, fatty acid β-oxidation (FAO) rate-limiting enzymes' mitochondrial Cpt1 and peroxisomal acyl-CoA oxidase 1 (Acox1) were induced to greater levels in the livers from GP73-high mice than in the livers from AAV-V

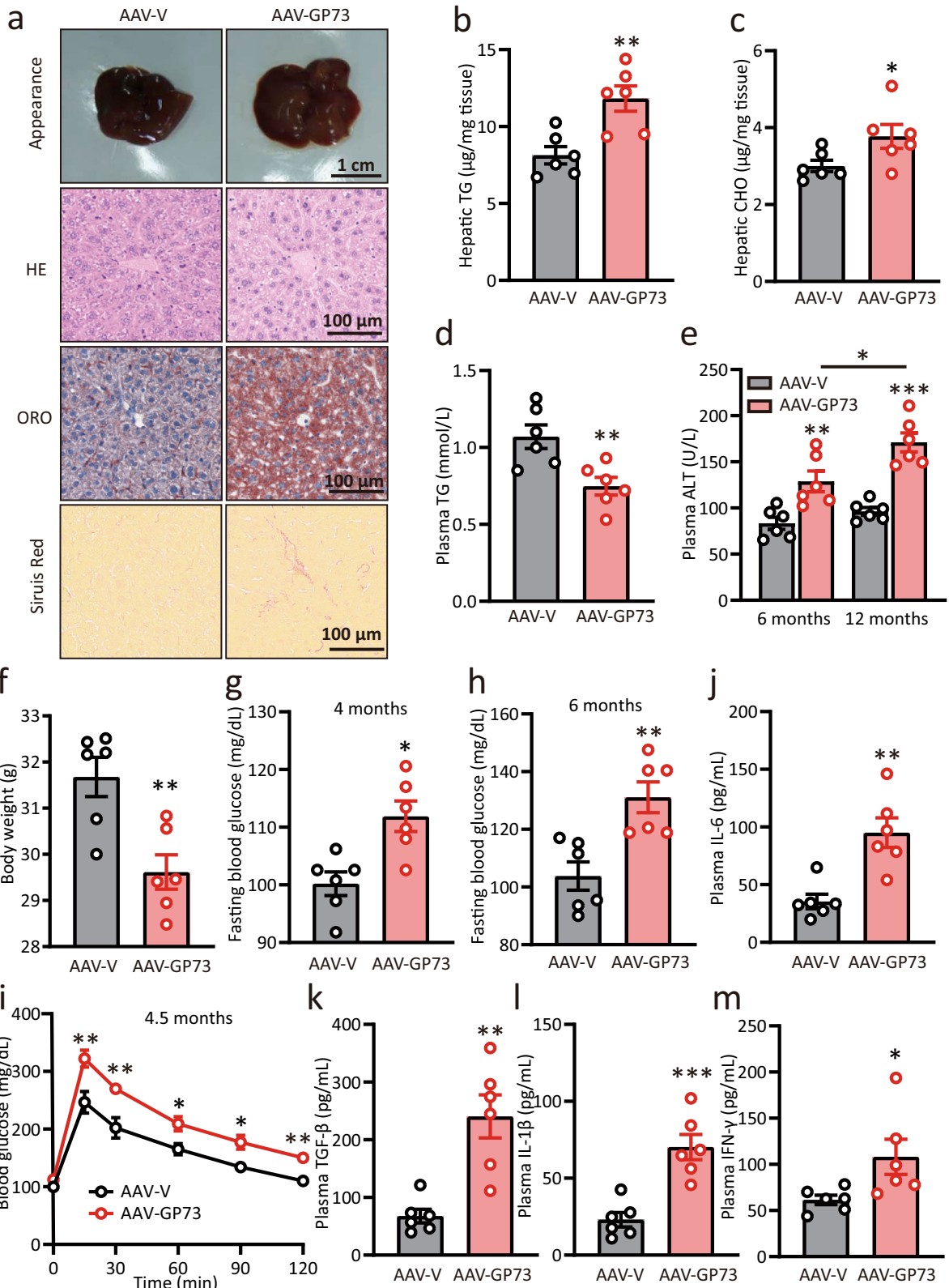

control mice (Fig. 4c and Supplementary Fig. 5). The FA bio-synthesis enzyme Fasn and the acetyl-CoA (substrate for lipid synthesis)-producing enzyme ACLY ATP-citrate lyase (Acly) were downregulated, whereas the cholesterol biosynthesis enzyme 3-hydroxy-3-methylglutaryl (HMG)-CoA synthase (Hmgcs) 2

and cholesterol-esterifying enzyme Acat1 were upregulated by GP73. Notably, increased expression of FA-uptake enzymes, including plasma membrane FA-binding proteins (Fabps) 1 and 2, was observed in the livers from GP73-high mice (Fig. 4c and Supplementary Fig. 5).

**Fig. 2 Chronic elevations in hepatocyte GP73 trigger non-obese NAFLD. a** Appearance, hematoxylin–eosin (HE), Oil Red O (ORO), and Sirius red staining of liver tissues from AAV-V- or AAV-GP73 ($3 \times 10^{11}$ vg)-injected mice fed a regular diet for 6 months. Data were repeated three times with similar results. **b–f** Hepatic levels of TGs (**b**) and CHO (c); plasma levels of TGs (**d**) and ALT (**e**); body weights (**f**) of mice injected with AAV-V or AAV-GP73 and fed a regular diet for 6 or 12 months ($n = 6$ per group). Differences between the two groups were evaluated using two-tailed Student's $t$-test. Data are presented as mean ± SEM. $^*P < 0.05$; $^{**}P < 0.01$. **g, h** Glucose levels in blood samples of 6 h-fasted AAV-V- or AAV-GP73-injected mice at 4 (**g**) and 6 (**h**) months after injection ($n = 6$ per group). Differences between the two groups were evaluated using two-tailed Student's $t$-test. Data are presented as mean ± SEM. $^*P < 0.05$; $^{**}P < 0.01$. **i** Glucose tolerance test (GTT) results for AAV-V- or AAV-GP73-injected mice at 4.5 months after injection ($n = 6$ per group). Differences between the two groups were evaluated using two-way ANOVA and Bonferroni's post hoc analysis. Data are presented as mean ± SEM. $^*P < 0.05$; $^{**}P < 0.01$. **j–m** Plasma levels of IL-6 (**j**), TGF-β (**k**), IL-1β (**l**), and IFN-γ (**m**) in AAV-V- or AAV-GP73-injected mice at 5 months after injection ($n = 6$ per group). Differences between the two groups were evaluated using two-tailed Student's $t$-test. Data are presented as mean ± SEM. $^*P < 0.05$; $^{**}P < 0.01$; $^{***}P < 0.001$.

A principal component analysis (PCA) showed that AAV-GP73 and AAV-V HFD were very close and partially overlapped in the PCA and were well separated from the AAV-V (Fig. 4d). Both groups shared over half (825) of the upregulated genes and 75% (590) of the downregulated genes (Supplementary Fig. 4c, d). KEGG pathway analysis of these overlapping genes mainly revealed enrichment in cellular metabolism pathways, including the TCA cycle, synthesis and degradation of ketone bodies, primary bile acid biosynthesis, FA metabolism, and PPAR signaling (Supplementary Fig. 4e). Of the top 50 highly upregulated genes and 50 highly downregulated genes affected by GP73, 90% were also affected by HFD, but with lower fold changes (Fig. 4e and Supplementary Fig. 6a, b). Of the unique genes affected by GP73, the hepatic expression of genes involved in FA metabolism (Hacl1, Cyp4a14, Thrsp, Mcat, Plin5, and Cyp4a10), cholesterol synthesis (Tm7sf2 and Insig), immune response (Nfil3 and Rnf125), AMPK signaling (Irs2 and Srebf1), and cell fate (Lrfn3, Ppm1k, Gadd45g, Arhgap28, Pstpip2, Spag17os, Fat2, and Hes6) was identified (Fig. 4f).

**GP73 promotes NASH progression in obese NAFLD.** After confirming the pathophysiological role of GP73 in non-obese NAFLD, we were interested in evaluating the role of GP73 in obese NAFLD. To this end, mice were fed a HFD for 12 months starting from week 4 after AAV-V or AAV-GP73 injection. Compared to AAV-V-injected mice fed with HFD, GP73-liver-high animals fed with HFD exhibited significantly increased weight gain, enlarged livers with visibly yellow appearances, increased liver-to-body weight ratio, and upregulated ApoB100 levels (Fig. 5a–d). Strikingly, liver-GP73-high mice developed steatosis, hepatocyte ballooning, and inflammatory cell infiltration sufficient to diagnose NASH (Fig. 5b). In addition, AAV-GP73-injected mice fed with HFD exhibited a higher degree of fibrosis (Fig. 5b), significantly increased hepatic lipid levels, and substantially elevated ALT and AST levels (Fig. 5e–h). These results indicate that GP73 accelerates the obesity-induced progression of steatosis to NASH and even to fibrosis.

**Metformin alleviates non-obese NAFLD induced by GP73.** Metformin is one of the first-line drugs for the treatment of type 2 diabetes[30]. However, the primary molecular mechanism of this biguanide is not well understood. Given the pathological effects of GP73 on hepatic lipids and glucose hemostasis, we wondered whether the GAP activity of GP73 is the target of metformin action. Indeed, metformin efficiently bound to endogenous GP73 and inhibited the interaction of GP73 with Rab23 (Fig. 6a, b). As a control, the interaction between TBC1D20 and Rab1b was unaffected by metformin treatment (Fig. 6c). We next performed an in vitro GAP assay using different concentrations of metformin. While the ability of TBC1D20-mediated Rab1b hydrolysis was unaffected by metformin, GP73-mediated catalysis of Rab23 GTP hydrolysis was inhibited by metformin in a dose-dependent

manner (Fig. 6d, e). Accordingly, the inhibitory effect of GP73 on ApoB and ApoB100 secretion was abolished by metformin (Fig. 6f, g).

Because livers with upregulated GP73 acquire the hallmarks of NAFLD, we investigated whether the administration of metformin by drinking water could mitigate the non-obese NAFLD phenotype induced by GP73 (Fig. 6h). We found that chronic metformin treatment reduced hepatic CHO and TG accumulation induced by GP73 overexpression (Fig. 6i, j). Importantly, metformin also prevented hyperglycemia in liver-GP73-high mice (Fig. 6k). Together, these results indicate that metformin improves the hyperglycemia and dyslipidemia induced by GP73 overexpression.

**GP73 expression is pathologically elevated in non-obese NAFLD mice and humans.** To establish the pathophysiological relevance of GP73 with non-obese NAFLD, we analyzed GP73 expression in multiple tissues from mice fed with high dietary cholesterol, a critical factor in inducing liver pathology associated with murine non-obese NAFLD. Compared with the results obtained with the regular-diet controls, we found strong upregulation of GP73 mRNA in the livers of mice fed a high-fat and high-cholesterol, cholate (HFHCC) diet for 1 month (Fig. 7a). In particular, GP73 expression was dramatically induced in livers from mice fed with HFHCC diet as early as day 7 after feeding (Fig. 7b). To continue the investigation, we measured the plasma concentrations of GP73 in 14 non-obese healthy controls and 14 NAFLD individuals without obesity diagnosed with MRI (Supplementary Table 1). A BMI less than 25 kg/m$^2$ was used to define the non-obese population. Compared with non-obese controls, patients with NAFLD without obesity had higher fasting glucose levels and a lower adiponectin-to-leptin ratio (Supplementary Table 1 and Fig. 7c–e). There were no significant differences between the two groups with regard to BMI, waist circumference, blood pressure, or platelets[31] (Supplementary Table 1). Notably, patients with NAFLD without obesity had significantly higher plasma GP73 levels than non-obese healthy subjects (Fig. 7f). Thus, dysregulated GP73 is associated with NAFLD without obesity.

**GP73 blockade improves whole-body metabolism in non-obese NAFLD mouse model.** To further explore whether inhibition of liver GP73 may improve metabolic disorders, we investigated the effect of sustained GP73 knockdown on metabolic homeostasis in mice with HFHCC diet-induced non-obese NAFLD. Mice were injected with siGP73 or control RNAi oligos twice a week during HFHCC diet feeding. After 4 weeks, approximately 50% knockdown of GP73 mRNA was found in the siGP73-treated group (Fig. 8a). As reported, the HFHCC diet recapitulated several characteristic features of non-obese NAFLD, including an increased liver-to-body weight ratio, upregulated hepatic CHO and TG levels, increased plasma AST and ALT levels along with elevated fasting glucose, and reduced insulin sensitivity[32]

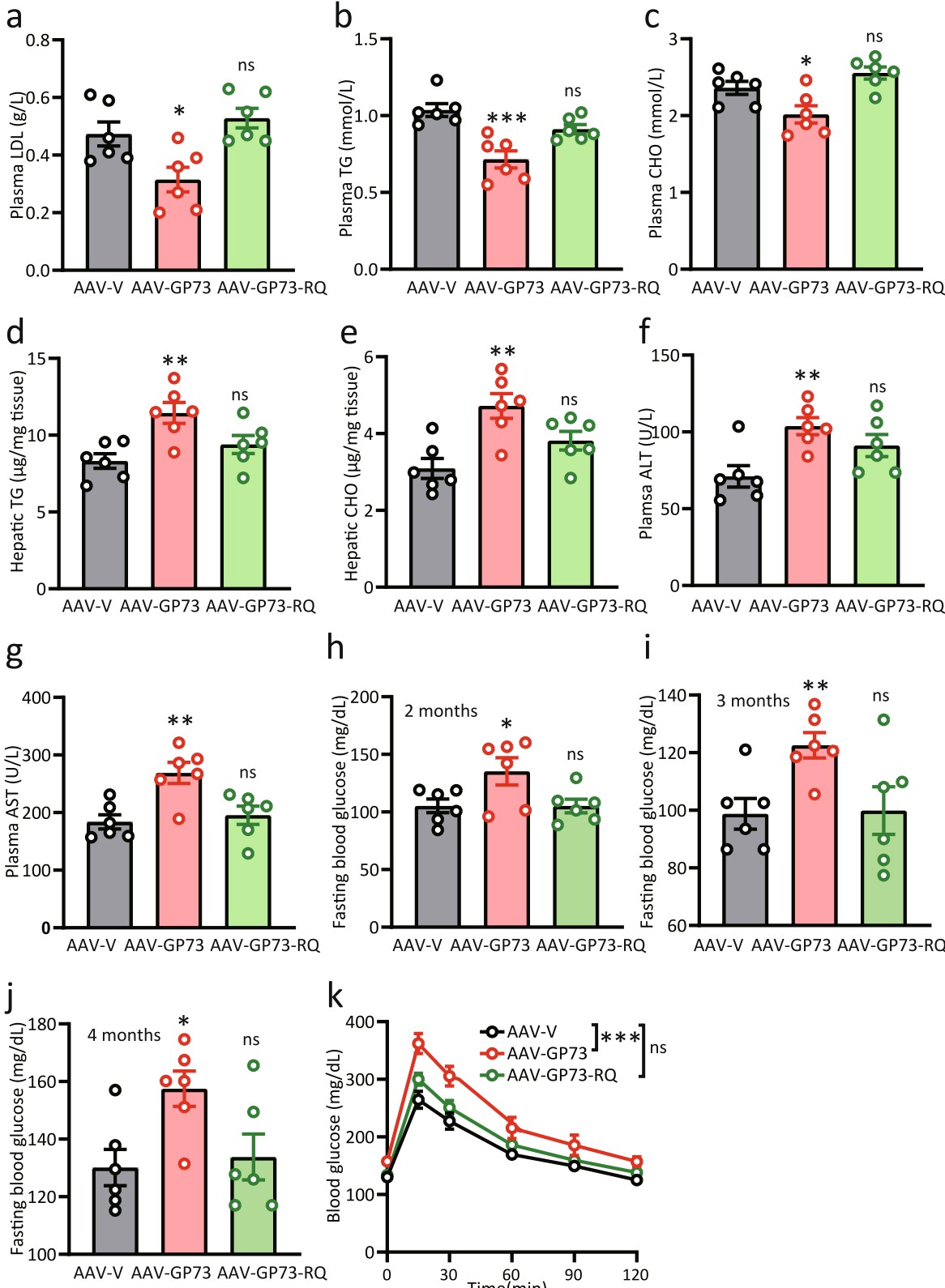

**Fig. 3 Induction of non-obese NAFLD by GP73 is highly dependent on its GAP activity. a–g** Plasma levels of LDL (**a**), TGs (**b**), CHO (**c**), ALT (**f**), and AST (**g**); hepatic levels of TGs (**d**), and CHO (**e**) in AAV-V-(Vector), AAV-GP73-(GP73), or AAV-GP73-RQ(GP73-RQ)-injected mice fed a regular diet for 4.5 months ($n = 6$ per group). Differences between the three groups were evaluated using one-way ANOVA and Bonferroni's post hoc analysis. Data are presented as mean ± SEM. ns, no statistical significance; $^{*}P < 0.05$; $^{**}P < 0.01$; $^{***}P < 0.001$. **h–j** Glucose levels in blood samples from 6 h-fasted AAV-V-, AAV-GP73-, or AAV-GP73-RQ-injected mice at 2 (**h**), 3 (**i**), and 4 (**j**) months after injection ($n = 6$ per group). Differences between the three groups were evaluated using one-way ANOVA and Bonferroni's post hoc analysis. Data are presented as mean ± SEM. ns, no statistical significance; $^{*}P < 0.05$; $^{**}P < 0.01$. **k** Glucose tolerance test (GTT) results for AAV-V-, AAV-GP73-, or AAV-GP73-RQ-injected mice at 4.5 months after injection ($n = 6$ per group). Differences between the three groups were evaluated using two-way ANOVA and Bonferroni's post hoc analysis. Data are presented as mean ± SEM. ns, no statistical significance; $^{***}P < 0.001$.

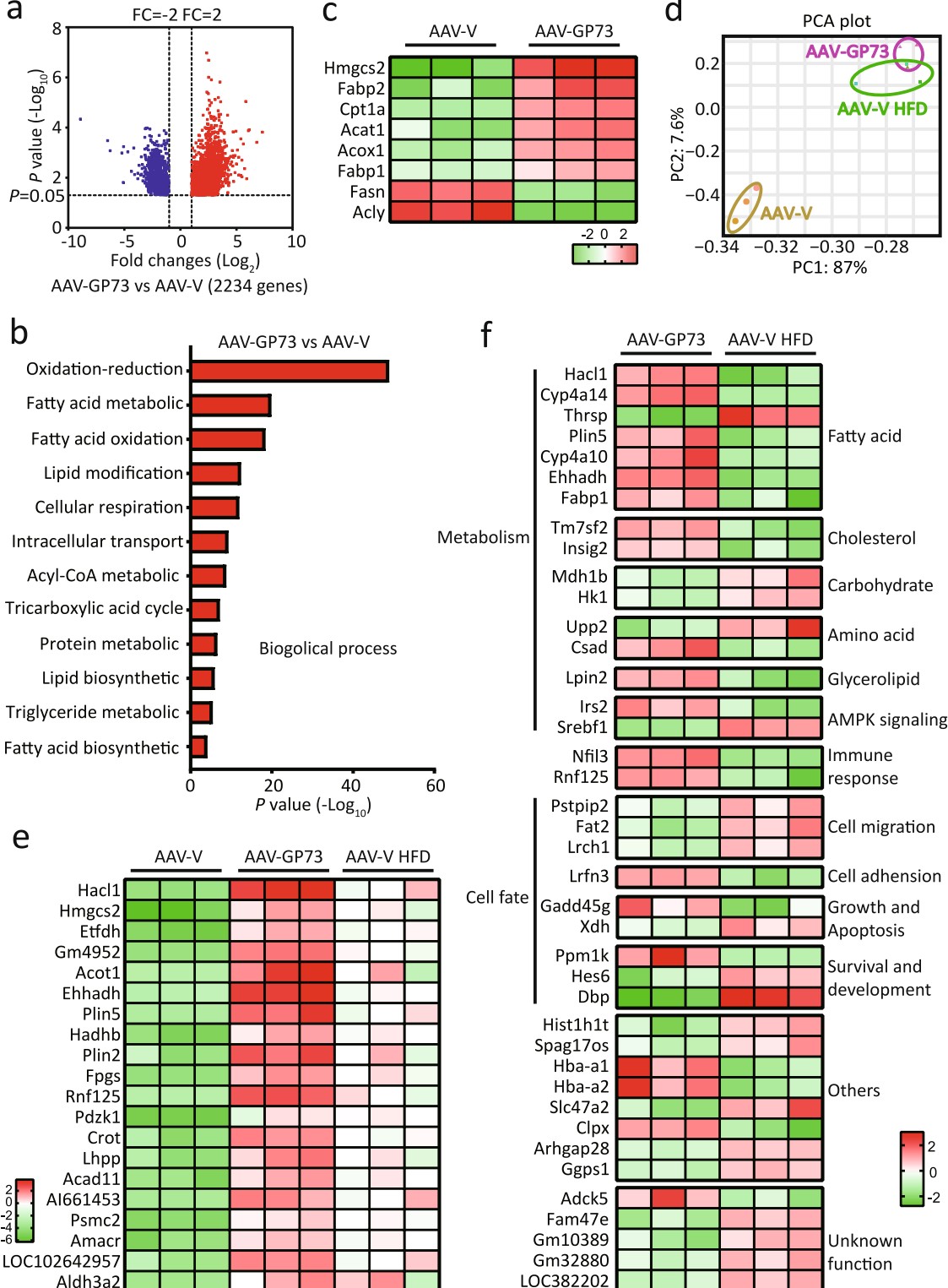

**Fig. 4 Gene expression signatures in non-obese NAFLD induced by GP73. a** Volcano plot of the DEGs in the livers from AAV-GP73-injected mice fed a regular diet for 12 months (*n* = 3 per group). Significantly downregulated genes are in blue, and significantly upregulated genes are in red. The data were analyzed with two-sided Student's *t*-test. The black vertical lines highlight fold changes (FCs) of −2 and 2, while the black horizontal line represents a *P* value of 0.05. **b** Pathways enriched for the DEGs in the livers from AAV-GP73-injected mice at 12 months after injection (*n* = 3 per group) according to GO term analysis at GO level 4. The bar plot shows significantly dysregulated pathways, and Fisher's exact test *P* values are shown on the *x*-axis. **c** Heatmap of the critical DEGs in the livers from AAV-GP73-injected mice versus AAV-V-injected mice fed a regular diet for 12 months. Upregulated genes are highlighted in red, and downregulated genes are highlighted in green. **d** PCA results. PCA was based on the gene expression patterns in the AAV-V (yellow), AAV-GP73 (purple), and AAV-V HFD (green) groups. In all plots, each point represents a sample. **e** Heatmap of the top 20 highly upregulated genes in the livers from the AAV-GP73 and HFD groups. **f** Heatmap of the top 20 highly upregulated and 20 highly downregulated genes that were specifically regulated by GP73.

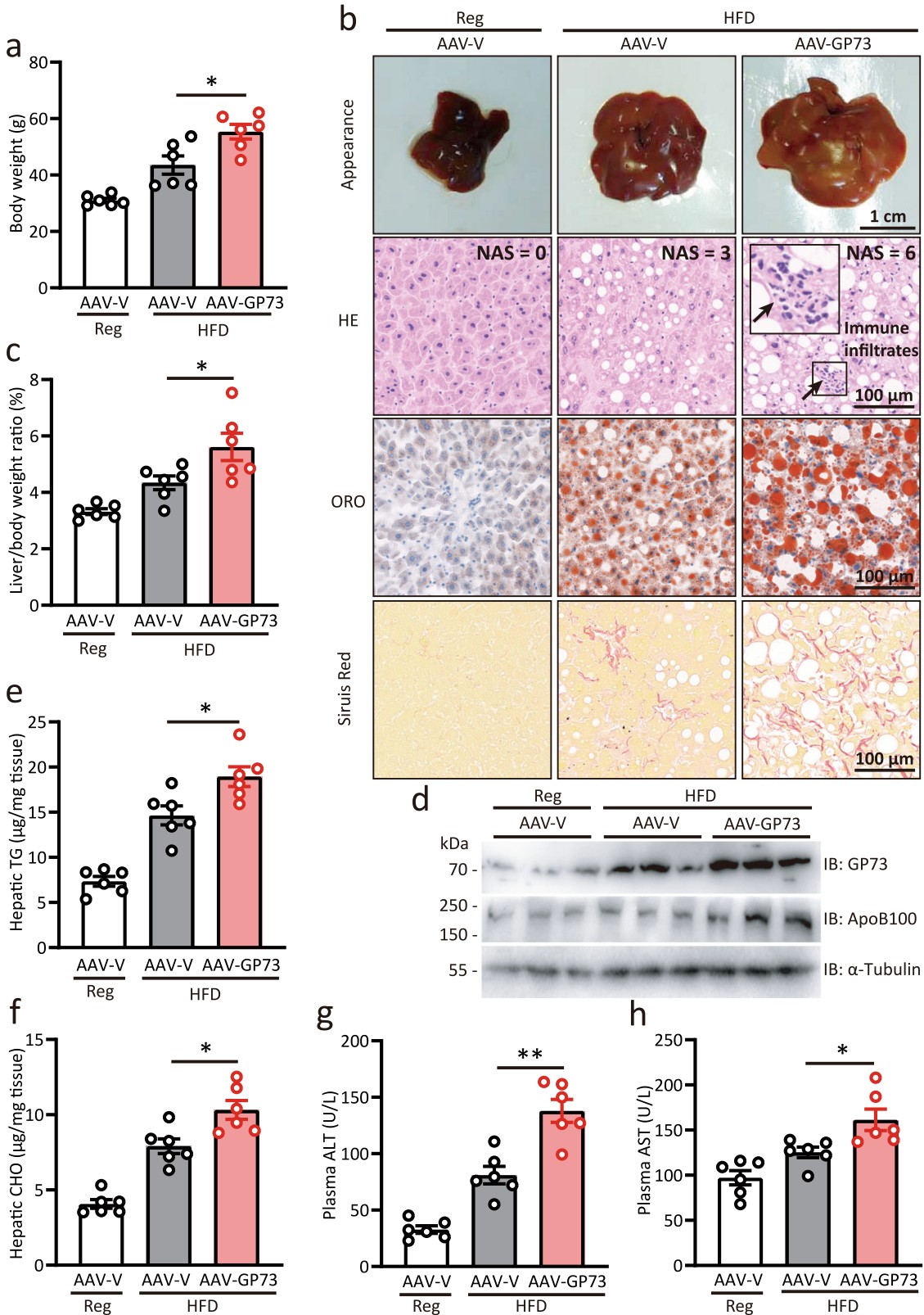

(Fig. 8b–h). The final body weight and inguinal white adipose tissue (WAT)-to-body weight ratio were unaffected (Fig. 8i, j). Of note, siGP73 administration alleviated liver damage and caused significant decreases in blood glucose levels (Fig. 8e–g). In the intervention protocol, knockdown of liver-GP73 also improved insulin sensitivity (Fig. 8h). In particular, hepatocyte TG and CHO levels were lower in the siGP73-treated group than in the control group (Fig. 8c, d). Thus, whole-body metabolism in non-obese NAFLD mice was significantly improved upon GP73 blockade.

## Discussion

As a type II transmembrane glycoprotein located in the Golgi, GP73 expression in hepatocytes is very limited or undetectable in

**Fig. 5 GP73 promotes NASH progression in obese NAFLD induced by HFD. a** Body weights of AAV-V-injected mice fed a regular diet and AAV-V- or AAV-GP73-injected mice fed a HFD for 12 months ($n = 6$ per group). Differences between the three groups were evaluated using one-way ANOVA and Bonferroni's post hoc analysis. Data are presented as mean ± SEM. $^*P < 0.05$. **b** Appearance, HE, ORO, and Sirius red staining of the livers from AAV-V-injected mice fed a regular diet and AAV-V- or AAV-GP73-injected mice fed a HFD for 12 months ($n = 6$ per group). **c** Liver-to-body weight ratio of AAV-V-injected mice fed a regular diet and AAV-V- or AAV-GP73-injected mice fed a HFD for 12 months ($n = 6$ per group). Differences between the three groups were evaluated using one-way ANOVA and Bonferroni's post hoc analysis. Data are presented as mean ± SEM. $^*P < 0.05$. **d** GP73 and ApoB100 protein levels in livers from AAV-V-injected mice fed a regular diet and AAV-V- or AAV-GP73-injected mice fed a HFD for 12 months ($n = 3$ per group). α-Tubulin was used as the equal loading control. Relative expression was calculated as the fold change in expression relative to the expression in No. 1 control mice. **e**–**h** Hepatic levels of TGs (**e**) and CHO (**f**); plasma levels of ALT (**g**), and AST (**h**) in AAV•V-injected mice fed a regular diet and AAV-V- or AAV-GP73-injected mice fed a HFD for 12 months ($n = 6$ per group). Differences between the three groups were evaluated using one-way ANOVA and Bonferroni's post hoc analysis. Data are presented as mean ± SEM. $^*P < 0.05$; $^{**}P < 0.01$.

healthy livers[33,34]. However, in patients with acute or chronic liver diseases, GP73 expression is significantly upregulated in hepatocytes[35,36]. We then wanted to assess the metabolic consequences of hepatic overexpression of GP73. After the consumption of a regular diet, mice with chronic elevation of GP73 in hepatocytes recapitulated almost all of the characteristic features of obese NAFLD, including reduced body weight, decreased plasma lipid levels, massive intrahepatic lipid accumulation, elevated baseline levels of inflammatory cytokines, and gradual insulin resistance development[3,4]. In addition, common and specific transcriptional changes in livers from GP73-induced non-obese NAFLD and diet-induced obese NAFLD were identified. Mechanistically, GP73 harbors potent TBC-domain GAP activity and was able to stimulate the GTPase activity of several Rab family members. The specific contribution of abnormal GP73 expression to the pathogenesis of non-obese NAFLD was largely dependent on the GAP activity of GP73. Strikingly, metformin could bind GP73 and inactivated the GAP activity of GP73. At clinically relevant plasma concentrations, metformin significantly ameliorated steatosis and improved insulin sensitivity in mice with GP73-induced non-obese NAFLD. By establishing a pathological role for elevated GP73 in non-obese NAFLD, our findings not only provide convincing evidence that elevated hepatic expression of GP73 can trigger non-obese NAFLD development in a GAP activity-dependent manner, but also provide a potent therapeutic target for blocking non-obese NAFLD development (Supplementary Fig. 7).

During secretion, VLDLs are transported from the ER to the Golgi in specialized transport vesicles. These ApoB-containing vesicles do not include ApoE or albumin despite the subsequent post-Golgi association between ApoE and VLDLs. ApoE may also be secreted on its own as a high-density lipoprotein (HDL). How these hepatic cargoes are secreted from living cells remains an open question. Here, we found marked reductions in VLDL-TG secretion in regular-diet-fed liver-GP73-high mice, which resulted in decreased steady-state plasma TG concentrations. Mechanistically, GP73 has the ability to stimulate the GTPase activity of several but not all of the Rab family members tested. Although GP73 showed the highest GAP activity toward Rab23, and the activity of Rab23 to promote the secretion of ApoB was in consistent with the phenotype that GP73 overexpression led to ApoB secretion impairment. Some other cargoes, including albumin, were stimulated both by Rab23 and GP73. Therefore, other Rabs tested or untested targeted by GP73 may be responsible for the observed differences. Regardless, our present provide further insight into the contribution of GP73 GAP activity on non-obese NAFLD, advancing our understanding of the physiological function of this important molecule.

Our gene expression microarray results indicated that cells respond to the intrahepatic accumulation of FAs by shutting down FA synthesis while simultaneously activating a parallel FAO pathway, as shown by increased expression of two rate-limiting enzymes in the mitochondrial and peroxisome FAO pathways and decreased expression of the FA biosynthesis enzyme FASN[37–39]. Since GP73 elevation induced excessive lipid accumulation even in the context of regular diet consumption, the source of these lipids is unclear. Increased FABP1 and FABP2 expression upregulated by GP73 were indicative of increased lipid uptake[40]. Moreover, the process of initiating cholesterol biosynthesis was increased, as two enzymes (ACAT and HMGCS) involved in the condensation of two and three molecules of the acetyl-CoA process were upregulated by GP73[41,42]. Our next question is related to whether many of these functions involve a common underlying mechanism. We proposed that GP73 functions in the etiology of non-obese NAFLD mainly through its GAP activity. Nonetheless, the contribution of GP73 GAP activity to obese NAFLD and the pathophysiological functions of GP73 independent of its GAP activity require further investigation.

Metformin is one of the first-line drugs for the treatment of type 2 diabetes and acts by inhibiting hepatic glucose production through AMPK-dependent and -independent mechanisms[43]. Direct inhibition of mitochondrial glycerol-3-phosphate dehydrogenase and mitochondrial respiratory chain complex 1 has been proposed for the acute inhibition of gluconeogenesis by metformin. Emerging evidence suggests that metformin could contribute to improvements in obesity-associated insulin sensitivity[44]. However, the host targets associated with the metabolic benefits of metformin remain incompletely understood. Here, we demonstrated that metformin can bind GP73, inactivate the GAP activity of GP73, and efficiently inhibit GP73-induced metabolic disorders at clinically relevant plasma concentrations. The findings of the GP73-mediated upregulation of AMPK expression and the glucogenesis signaling pathway support this result. Collectively, our data illustrate a previously unidentified mechanism of action for metformin.

The evolutionary basis for the actions of GP73 described here is of interest. We proposed that during acute stress or danger, GP73 tends to increase FA uptake and cholesterol synthesis, which results in the conversion of incoming FFAs and cholesterol into lipid drops or VLDLs for storage. Metabolic rewiring may help these cells cope with external danger or regulate the antiviral signaling immune response. Chronic elevation of GP73 in hepatocytes leads to persistent impairment of metabolic homeostasis, driving insulin resistance and non-obese hepatic steatosis. The increase in hepatocyte GP73 may be induced by chronic infection, inflammation, cholesterol, or unknown factors. Regardless of the stimuli, chronic upregulation of GP73 in hepatocytes contributes to the early onset of non-obese NAFLD. A Western diet combined with pathologically high GP73 levels can promote the progression from steatosis to NASH. This finding may provide a potent therapeutic target for blocking non-obese NAFLD development and disease progression.

## Methods

**Study subjects.** The Committee for Ethics in Human Studies from the Third Medical Center of the Chinese PLA General Hospital approved this study (KY2021–009) and abided by the Declaration of Helsinki principles. Fourteen non-

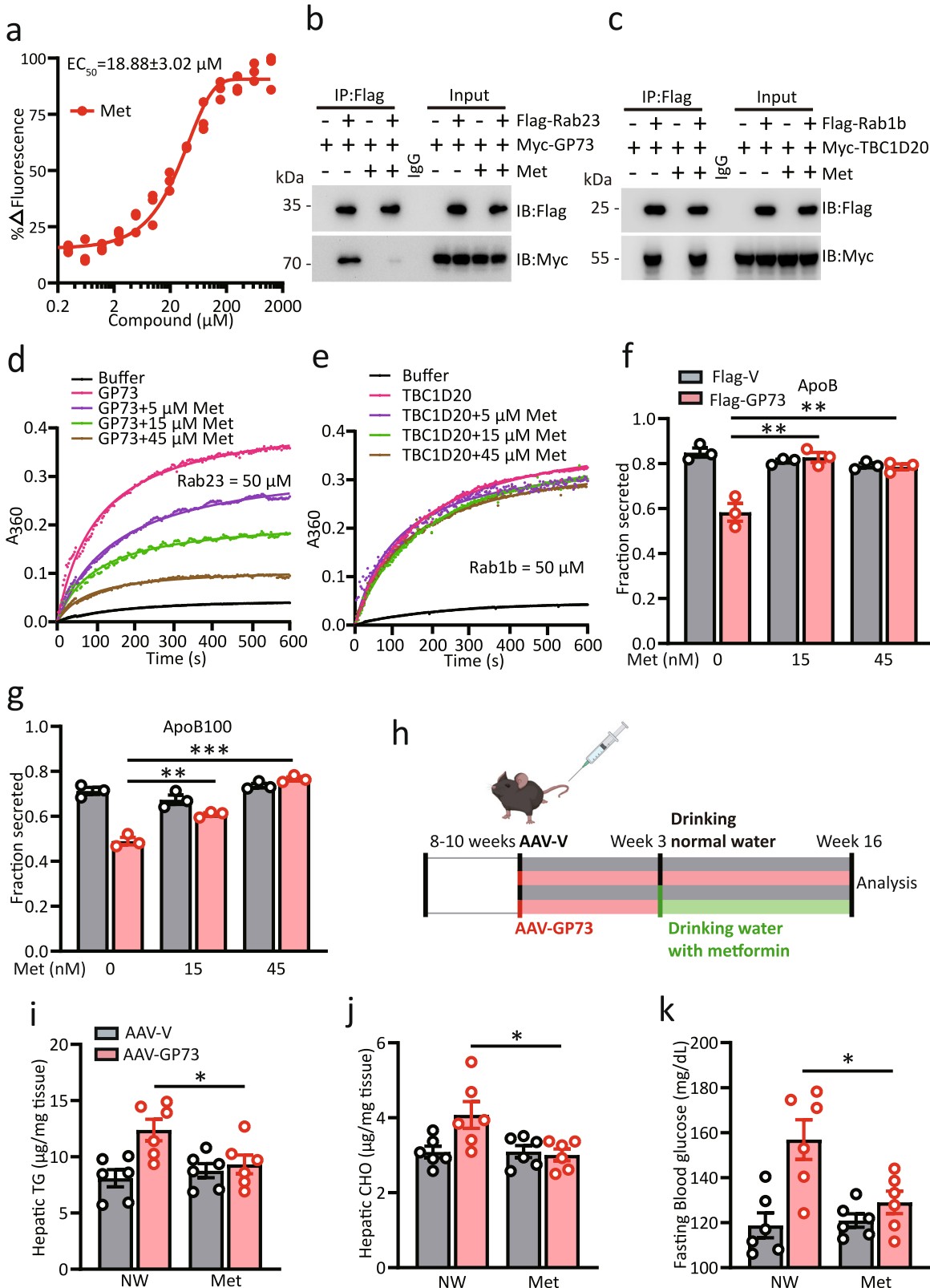

obese healthy subjects and 14 NAFLD subjects without obesity diagnosed with MRI were included in the study. NAFLD with <25 kg/m² BMI was defined as NAFLD without obesity. All individuals in this study provided a signed statement of informed consent. All patients had no history of the use of any hepatotoxic drugs, hormone replacement therapy, or herbal products and consumed no more than 20 g/day alcohol. The healthy control group had no illnesses, no use of alcohol, drugs, or herbal substances, no history of previous liver diseases and was negative for viral hepatitis serology tests.

**Clinical assessment**. Each participant was required to complete a questionnaire for self-reported alcohol consumption, smoking and past medical history. Anthropometric data were acquired by two specialized doctors and included weight, height, waist circumference (WC), systolic blood pressure (SBP), and diastolic blood pressure (DBP). BMI was calculated as weight divided by the height squared (kg/m²). Non-obese was determined by a threshold of <25 kg/m². After fasting overnight for ≥8 h, blood samples were taken for routine blood tests and

**Fig. 6 Metformin alleviates non-obese NAFLD induced by GP73. a** Microscale thermophoresis (MST) analysis of the interaction between metformin (Met) and GP73. The data were derived from the effect of metformin on the fluorescence decay of fluorescently labeled GP73. The half-maximum effective concentration (EC$_{50}$) was determined by the Hill slope. $n = 3$ independent biological experiments. **b, c** Immunoprecipitation analysis of the interaction between GP73 and Rab23 (**b**) or TBC1D20 and Rab1b (**c**) in the presence or absence of Met. Data were repeated three times with similar results. **d, e** Kinetic analysis of GP73 activity toward Rab23 (**d**) or TBC1D20 activity toward Rab1b (**e**) in the presence of different concentrations of Met. $n = 3$ independent biological experiments. **f, g** ApoB (**f**) and ApoB100 (**g**) secretion efficiency in cells from Flag-vector- or Flag-GP73-transfected cells treated with Met. $n = 3$ independent biological experiments. Differences between the two groups were evaluated using one-way ANOVA and Bonferroni's post hoc analysis. Data are presented as mean ± SEM. **P < 0.01; ***P < 0.001. **h** Schematic depicting the experimental setup. **i–k** Hepatic levels of TGs (**i**) and CHO (**j**); 6 h-fasted glucose levels (**k**) of AAV-V- or AAV-GP73-treated mice given normal drinking water (NW) or drinking water with metformin (3 g/L) (NW) for 4 months ($n = 6$ per group). Differences between the four groups were evaluated using one-way ANOVA and Bonferroni's post hoc analysis. Data are presented as mean ± SEM. *P < 0.05.

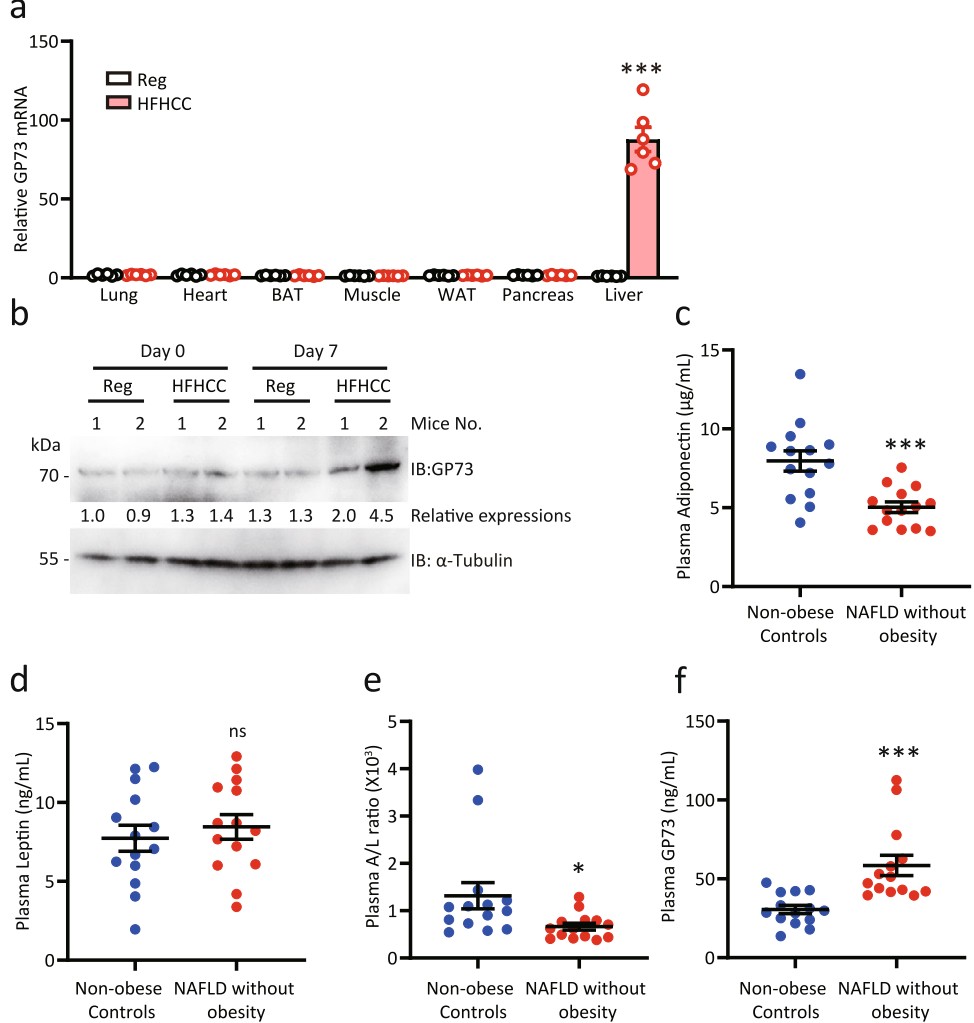

**Fig. 7 GP73 expression is pathologically elevated in NAFLD patients without obesity. a** GP73 mRNA expression in multiple tissues from mice fed a regular diet (Reg) or a high-cholesterol, high-cholate diet (HFHCC) for 1 month ($n = 6$ per group). Differences between the two groups were evaluated using two-sided unpaired Student's $t$-test. Data are presented as mean ± SEM. ***P < 0.001. **b** GP73 protein levels in the livers from mice fed HFHCC for 7 days ($n = 2$ per group). α-Tubulin was used as the equal loading control. Relative expression was calculated as the fold change in expression relative to the expression in No. 1 control mice. **c–f** Plasma adiponectin (**c**), leptin (**d**), and adiponectin-to-leptin ratio (A/L ratio; **e**); GP73 levels (**f**) in NAFLD patients without obesity and non-obese healthy subjects ($n = 14$ per group). Differences between the two groups were evaluated using the two-sided unpaired Student's $t$-test. Data are presented as mean ± SEM. ns, no statistical significance; *P < 0.05; ***P < 0.001.

GP73 level assays using ELISA kits for human GP73 (no. 03.03.0201, Hotgen Biotech Co., Ltd. Beijing, China).

**Reagents**. Metformin (PHR1084) and tyloxapol (T8761) were purchased from Sigma-Aldrich (MO, USA). Glucose (20171108) was purchased from Sinopharm Chemical Reagent Co., Ltd. (Beijing, China). A blood glucose meter (06656919032)

and test strips (1072332990) were purchased from Roche (Basel, Switzerland). Dulbecco's modified Eagle's medium (DMEM, high-glucose, D5796) was purchased from Millipore (MA, USA). Lipofectamine 2000 (11668027) was purchased from Invitrogen (Carlsbad, CA, USA). PerfectStart$^{TM}$ Green qPCR SuperMix (AQ601) and TransScript® One-Step gDNA Removal and cDNA Synthesis SuperMix (AT311) were purchased from TransGen Biotech (Beijing, China). Biochemical kits for mouse AST (200218), ALT (191230), TG (200224) and CHO

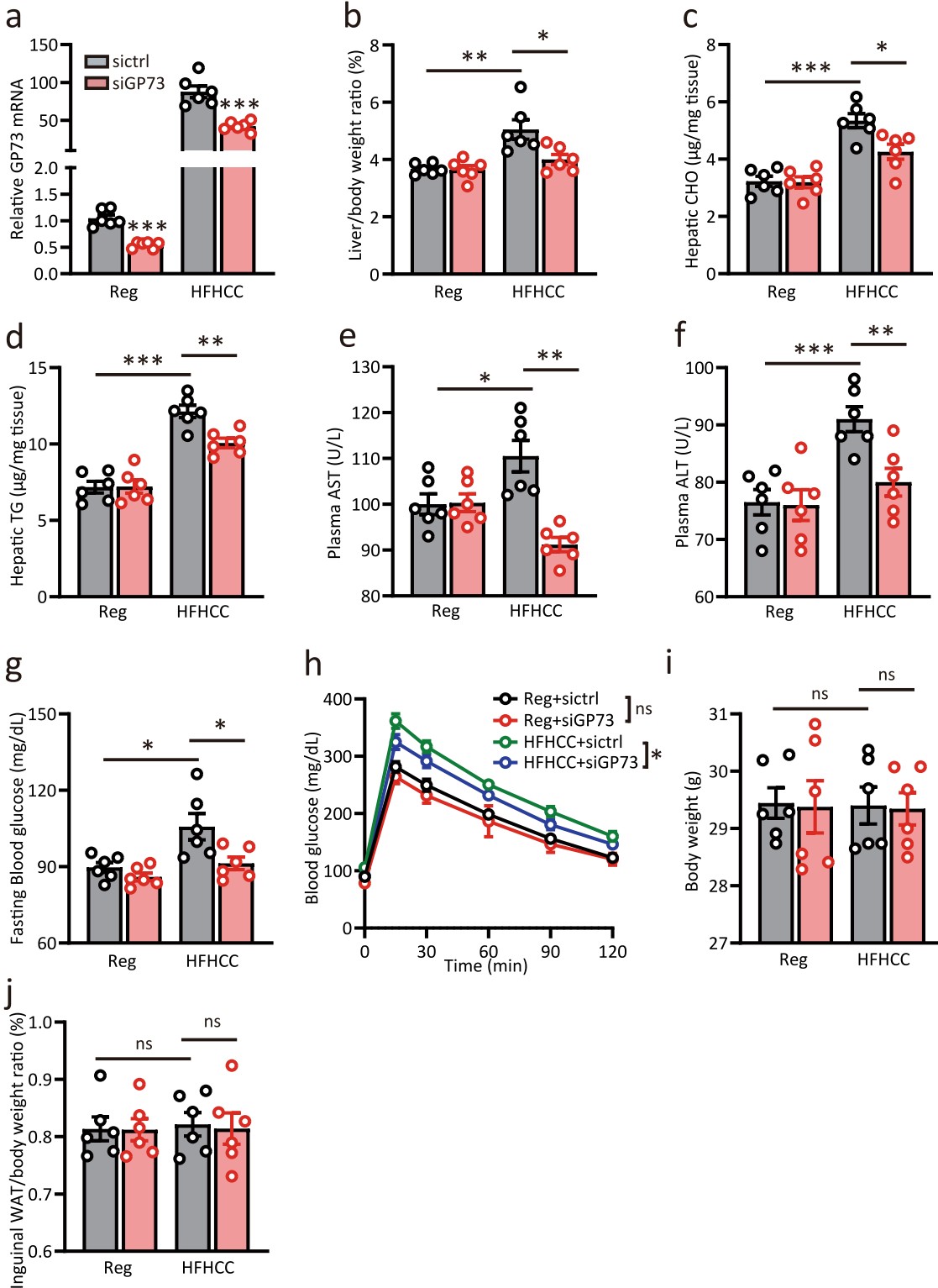

**Fig. 8 GP73 blockade improves whole-body metabolism in non-obese NAFLD mouse model. a** GP73 mRNA expression in livers from mice fed a regular or HFHCC diet for 1 month and injected with siGP73 or control RNAi oligos twice a week ($n = 6$ per group). Differences between the two groups were evaluated using unpaired Student's $t$-test. Data are presented as mean ± SEM. ***$P < 0.001$. **b–j** Liver-to-body weight ratio (**b**); hepatic levels of CHO (**c**) and TGs (**d**); plasma levels of AST (**e**) and ALT (**f**); fasting glucose levels (**g**) and GTT (**h**); body weight (**i**) and inguinal WAT-to-body weight ratio (**j**) in the four groups above ($n = 6$ per group). Differences between two groups were evaluated using unpaired Student's $t$-test. Data are presented as mean ± SEM. ns, no statistical significance; *$P < 0.05$; **$P < 0.01$; ***$P < 0.001$.

(200224) were purchased from Ruierda Biological Technology (Beijing, China). ELISA kits for mouse ApoB (SEC003Mu), ApoB100 (PAA603Mu01), albumin (CEB028Mu), ApoE (SEA704Mu), ApoA1 (SEA519Mu), TG (CEB687Ge), and CHO (CEB701Ge) were purchased from Cloud-Clone Corp. (Wuhan, China). ELISA kits for mouse TGF-β (121702), IFN-γ (120062), IL-1β (1210122), and IL-6 (1210602) were purchased from Dakewe Biotech Co., Ltd. (Beijing, China). ELISA kits for human leptin (SEA084Hu) and adiponectin (SEA605Hu) were from Cloud-Clone Corp (Wuhan, China).

**Antibodies**. The following antibodies were used: anti-α-tubulin for western blotting (Proteintech, cat: 11224-1-AP, lot: 00084996, 1:5000 dilution); anti-GP73 for western blotting (Abcam, cat: ab92612, clone: OTI6C9, lot: GR3250053-6, 1:2000 dilution); anti-ApoB for western blotting (Proteintech Group, cat: 20578-1-AP, lot: 00025612, 1:1000 dilution); anti-mouse Flag for western blotting (Sigma-Aldrich, cat: F1804, lot: SLBF6631, 1:5000 dilution); anti-mouse Myc for western blotting (ABclonal, cat: AE010, lot: 4000049011, 1:2000 dilution); anti-GP73 for immunofluorescence (BOS-TER, cat: A02975-2, lot: BOS7542BP4109, 1:100 dilution); and anti-albumin for immunofluorescence (ABclonal, cat: A0353, lot: 0202160201, 1:100 dilution).

**Plasmids**. Mammalian expression vectors encoding N-terminal Myc-tagged and C-terminal Flag-tagged GP73 were constructed by inserting the corresponding PCR-amplified fragments into pcDNA3.1 (V79520, Invitrogen). pcDNA3.1-Flag-GP73-RQ was constructed with a Fast Site-Directed Mutagenesis Kit (Tiangen, Beijing, China). All constructs were sequence verified.

**Cell culture and transfection**. The Huh-7 (0403) cell line was obtained from the Japanese Collection of Research Bioresources and had been previously tested for mycoplasma contamination. Huh-7 cells were incubated in DMEM at 37 °C under a humidified atmosphere with 5% $CO_2$. All media were supplemented with 10% fetal bovine serum (FBS), 100 U/mL penicillin, 0.1 mg/mL streptomycin, 1× non-essential amino acid solution, and 10 mM sodium pyruvate. Lipofectamine 2000 (Invitrogen) was used for transfection following the manufacturer's protocol.

**Recombinant proteins**. Recombinant GP73 protein with a 6x His tag was produced in 293T cells and purified with a B-PER 6x His Spin Purification Kit before endotoxin removal with a commercial system (Lonza, Inc.). Recombinant Rab protein with a pCDNA3.1-Flag tag was produced in 293T cells and purified with a pCDNA3.1-Flag Spin Purification Kit before endotoxin removal with a commercial system (Lonza, Inc.). The proteins were concentrated using Amicon ultracentrifuge filter devices (Millipore). The protein concentrations were determined using a Bradford assay (Bio-Rad), and protein purity was examined by Coomassie blue staining of SDS-PAGE gels.

**Immunoblot analysis**. The tissues and cells used for immunoblot analysis were lysed in buffer containing 150 mM NaCl, 50 mM Tris-HCl pH 7.5, 0.1% w/v SDS, 0.5% w/v Na-deoxycholate, 1% v/v Nonidet P-40, 1 mM ethylenediaminetetraacetic acid (EDTA), 1 mM ethylene glycol-bis(β-aminoethyl ether)-N,N,N′,N′-tetraacetic acid (EGTA), 2.5 mM Na-pyrophosphate, 1 mM NaVO4, 10 mM NaF, and fresh protease inhibitors from Sigma. The lysates were centrifuged at 4 °C for 15 min at 13,000g, and the proteins in the supernatants were separated by SDS-PAGE after centrifugation. The separated proteins were transferred to polyvinylidene fluoride (PVDF) membranes for immunoblot analyses using the indicated primary antibodies. Anti-rabbit or anti-mouse HRP-conjugated antibodies were then applied after three washes and the antigen–antibody complexes were visualized using chemiluminescence by a Tanon 5200 chemiluminescent imaging system, and analyzed by Tanon Gelcap (v.5.22). The band intensities representing the indicated relative expression levels were quantitated with reference to the α-tubulin band intensities and the levels in the control group.

**Secretion assays**. Cells were transfected with Flag-vector or Flag-GP73, and 24 or 48 h later, the cells were washed and allowed to secrete cargo into fresh medium for 6 h. The medium and cell lysates were collected, and the cargo content was measured by ELISA. Secretion efficiency was calculated as the fraction secreted, defined as the ratio between the amounts of cargo that was secreted and the total amount of cargo (secreted plus cell-associated cargo) present in a well.

**Animals**. Eight-week-old male C57BL/6N WT mice were purchased from SPF Biotechnology (Beijing, China). Mice were maintained in a pathogen-free, temperature-controlled, and 12 h light and dark cycle environment with the temperature maintained at 21–23 °C, relative humidity at 50–60%, and free access to food and water at the AMMS Animal Center (Beijing, China). All mice were group-housed for 3 days before any procedures. All animal experiments were conducted at the AMMS Animal Center (Beijing, China) and were approved by the Institutional Animal Care and Use Committee and the studies have complied with all relevant ethical regulations regarding the use of research animals.

For AAV-GP73 injections, AAVs encoding mouse GP73, GP73 R248K and the Q310A mutant were constructed and propagated by Hanbio, Inc., Shanghai, China. The mice were injected with $3 \times 10^{11}$ vg intravenously every 6 months.

For VLDLs' secretion, the mice were fasted overnight and then intravenously injected with tyloxapol (400 mg/kg body weight). Blood was collected at the indicated time points, and plasma TG levels were determined according to the relevant kit manufacturer's instructions.

For glucose tolerance tests, the mice were fasted for 6 h. Fasting blood glucose levels were measured in blood collected from the tail vein by tail snipping, and then, glucose (1.5 g/kg body weight) was administered intravenously to conscious animals. Glucose was measured in blood collected from the tail vein at 0, 30, 60, 90, and 120 min post injection.

To establish a HFD-induced NAFLD model, male C57BL/6N mice were maintained on a regular chow diet (Teklad 2919, 9% fat) or HFD (Teklad TD.06414, 34.3% fat) for 6 months. To establish a HFHCC-induced non-obese NAFLD model, male C57BL/6N mice were maintained on a regular chow diet or fed a HFHCC diet (Teklad TD.90221, 15.8% fat, 1.25% cholesterol, and 0.5% sodium cholate) for 4 weeks. All diets were purchased sterile (irradiated) and stored as per recommendations from the manufacturer. Feed provided to mice was changed every 3 days. During this time, GP73 or control siRNA oligos were administered at a dose of 0.2 nmol/g to mice via tail vein injection twice a week. Mice were then sacrificed for the collection of plasma and liver tissues after 4 weeks of feeding. Samples were snap frozen in liquid nitrogen and stored at −80 °C until analysis.

**Chemical modifications siRNA for in vivo study**. In the sense strand of siRNA, the 3′ terminus was modified by cholesterol and 4 thiols, the 5′ terminus was modified by 2 thiols, and the whole strand was modified by 2′-O-methyl–modified (2′ OMe-modified). All siRNAs were obtained from GenePharma.

The siRNAs targeting GP73 had the following target sequences:
siGP73 (275): 5′- CCUGGUGGCCUGUGUUAUUTT −3′
Scrambled siRNA oligonucleotides from siGP73 were used as a control (siCtrl).

**Quantification of glycemia, plasma metabolites, cytokines, and hepatic lipids**. Glycemia was measured with whole blood collected from the tail vein using a Roche blood glucose meter. For measurement of fasting blood glucose, fasting was started at 9:00 a.m., and the mice were fasted for 6 h. Random blood glucose levels were also measured at 9:00 a.m. If the glucose level was greater than 630 mg/dL (the upper detection limit of the glucometer), a value of 630 mg/dL was recorded. The blood glucose levels were determined.

Plasma was separated using lithium heparin-coated microcentrifuge tubes (BD Diagnostics). Plasma LDL, TG, CHO, ALT, AST, and cytokine levels were measured using kits according to the manufacturers' instructions. For the hepatic lipid assay, approximately 100 mg of liver tissue was homogenized in methanol, and lipids were extracted in chloroform: methanol (2:1 v/v) for 12 h. Hepatic TG and CHO levels were then quantified using mouse TG and CHO ELISA kits.

**qRT-PCR**. First-strand cDNA was synthesized using a TransScript One-Step gDNA Removal and cDNA Synthesis SuperMix Kit (Transgen, AT311-03). qRT-PCR was performed in the QuantStudio 3 Real-Time PCR System (Applied Biosystems) using SYBR Green PCR Master Mix (Transgen, AQ601-04). Each sample was analyzed in triplicate with GAPDH as the internal control.

The primer sequences are as follows:
GP73: 5′-CGTCGCAGCATGAAGTCTC-3′ (forward), 5′-CAGTAGTTGAAG CCTAGCACAAT-3′ (reverse);
GAPDH: 5′- AGGTCGGTGTGAACGGATTTG −3′ (forward), 5′-GGGGTC GTTGATGGCAACA-3′ (reverse)

**Statistics**. In mice experiment, no statistical methods were used to predetermine sample size. We used $n = 6$ in each group based on prior publications using comparable methods[45–47]. For this scenario, the majority of the power values for mice studies were over 0.8. For the human study, based on our preliminary data, differences in plasma GP73 levels between healthy peoples and NAFLD without obesity are ≥150% with SEM within groups of ≤20%. Hence, setting the significance level α to be 0.05 and the effect size ≥1.4, the minimum sample size required for a Power (1−β) of 0.8 was calculated to be ≥10 subjects per group. The data are presented as the mean ± SEM or as the median (interquartile range). Differences between two independent samples were evaluated by two-tailed Student's $t$-test. Differences among multiple samples were analyzed by one-way or two-way ANOVA followed by Bonferroni's post hoc analysis. The significance values were set as follows: ns (not significant), $P > 0.05$; $^*P < 0.05$; $^{**}P < 0.01$; and $^{***}P < 0.001$. Statistical analysis was performed using GraphPad Prism 8.0.

**Reporting summary**. Further information on research design is available in the Nature Research Reporting Summary linked to this article.

## Data availability

Gene expression microarray data generated for this manuscript have been deposited in the Gene Expression Omnibus database under accession number GSE163918. The data that support the findings of this study are available from the corresponding author. Source data are provided with this paper.

## Code availability

The code used for automated analysis and fitting is available on reasonable requests from the corresponding authors.

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

## Acknowledgements

We thank the National Key Research and Development Program of China (grant no. 2018YFA0900800), the National Natural Science Foundation of China (grant nos. 32070755, 82070595, 81972696, 81671973, 81860502, 31670761, 31872715, and 81773205) and the Project funded by China Postdoctoral Science Foundation (grant nos. 2020M683743) for their support.

## Author contributions

F.Y., Q.G., X.L.Y., F.X.W., H.Z. and C.W.W. designed the experiments. C.W.W., H.Z., L.M.W., Y.M.P., Q.Z., C.Q.L., Z.W.S., M.Y.L., G.C., E.H.M. and H.L.L. collected and analyzed the data. J.Y.F., Y.M.P., Q.Z., L.M.W., G.C., S.Q.D., E.H.M., H.L.L., X.P.Y., H.T.L., L.F. H., Y.X.X. and Y.H. Z. carried out mice assays. Y.M.P., L.M.W., H.L.L., J.F. L., J.L., X.M. Z., J.L. L., D.R.L., M.W., M.Y.L., and Y.H.M. carried out cell lines experiments. X.T.M., X.W.D., D.D.Z., H.Q.H. and X.L.Y. collected clinical data. X.P.Y., Q.Z., E.H.M., L.M.W., C.W.W. and H.Z. analyzed the data and prepared the manuscript.

## Competing interests

The authors declare no competing interests.
