## [Peer Review File. · Nature Communications]

Reviewers' Comments:

Reviewer #1:

Remarks to the Author:

In this study the authors assessed the role of GP73 in the development of non-obese NAFLD in patients with non-obese NAFLD and mouse models of non-obese NAFLD. Key findings include that mice overexpression GP73 showed signs of NAFLD and that patients with non-obese NAFLD showed elevated levels of GP73. While the authors present a lot of data there are some issues that need to be addressed:

1. Please provide more details on species and experimental set up used in the study in the abstract.
2. There are several statements in the introduction that lack references e.g., dietary pattern of lean patients with NAFLD, microbiota in lean NAFLD patients. Indeed, there are by now several studies in mice that show that the dietary composition e.g., the amount and kind of fat as well as sugar might be critical. The same of intestinal barrier dysfunction and LPS shown to also affect lipid export.
3. Metformin is a drug affecting many pathways, starting with intestinal microbiota, intestinal barrier, glucose metabolism in liver, PAI-1 expression... maybe another drug being more specific would be a better choice for a proof of concept. The same for berberine, where molecular mechanisms of action are not yet fully understood.
4. Please provide methods in the main body of the manuscript.
5. Please provide information of power-calculation for both human and mouse studies.
6. Ultrasound is not a very reliable way to stage NAFLD, the same fore a self-reported assessment of alcohol intake etc.
7. Was fasting of mice started a 9 o'clock or at 3 o'clock at night?
8. How specific the overexpression of GP73 for the liver?
9. Please provide information on visceral adipose tissue content in patients and controls (ratio of adiponectin and leptin, MRIs or something alike).
10. How specific is the overexpression of GP73 in the mice treated with the adenovirus in hepatocytes? How about other liver cells and tissues?
11. The authors show ApoB100, what about ApoB48?
12. The description of the animal procedures is rather unclear in the methods section.
13. How was food intake of mice with liver-GP-73 high? Also, how was body weight in older mice? Did animals continue to gain less weight or lose weight?
14. Why were different ages of mice used? Also, did liver damage progress? Did animals develop at older age and prolonged overexpression of GP73 develop fibrosis?
15. The comparison to HFD is not adequate as these mice normally become overweight and therefore metabolism may markedly differ. Also, after 12 months HFD fed mice should have beginning fibrosis. An n of 3 is very low.
16. GP73 expression in human liver, not only plasma should be presented as the origin of GP73 cannot be distinguished this way. Also, it seems that levels vary considerable suggesting that other factors might be more important in these patients.
17. A group of mice only receiving metformin and not treated with AAV-GP73 should be shown.
18. As the authors mention the role of microbiota and intestinal barrier in the introduction and the liver and gut have been shown to interact, did the treatment with AAV-GP73 somehow affect intestinal barrier function?

Reviewer #2:

Remarks to the Author:

The present study investigates the role of GP73 in the pathogenesis of NAFLD in a mouse model. Data include lipid analysis either by enzymatic testing and lipidomic workflows. Based on my expertise I will specifically comment on lipidomics data. Unfortunately, the lipidomic methods are not described in sufficient detail. However, based on the current description, the applied method does not meet current state of the art for lipid species quantification.

- Samples were extracted including methanol, MTBE and internal standard mixture. Suitable lipid extraction is key for accurate lipid quantification. Please add details of the procedure including sample and solvent volumes and composition of the internal standard mixture.

- It is unclear, how data were measured (which mass transitions; were isomeric and isobaric interferences considered; how was quantification performed etc.). Please add these details and suppl. tables including mass transition, RT, internal standard for all measured lipid species.
- Was the method including lipid extraction validated? If yes, please include references or data concerning reproducibility, LOD etc.?
- Figure S3 shows a heat map of regulated lipid species. Lipid species and molecular lipid species (i.e. acyl chains) were annotated. I doubt that sn-positions are proven by experimental data. Please use the hierarchical concept and annotation of the latest shorthand nomenclature for lipid species (DOI: 10.1194/jlr.S120001025). Please describe how fold changes were calculated (based on counts or nmol?). In part annotations are not correct e.g. PC 42:2 (O-20:1/22:1) should read PC O-42:2.
- Line 232: "Lipid levels were normalized to total lipid contents and transformed as log2 fold-change relative to the AAV-V control." This kind of normalization most likely leads to wrong conclusions due to substantially different TG content of the samples. Consequently, subsequent data interpretation is presumably not valid. Comparison should be based on molar concentrations without normalization to total lipid content (especially when samples show a high degree of lipid storage). Most likely, downregulated PC and PE fraction (Fig. 4E) is derived from normalization – I guess related to wet weight or protein these phospholipid classes remain unchanged.

Reviewer #3:

Remarks to the Author:

This manuscript by Peng et al. potentially establishes the molecular basis of non-obese non-alcoholic fatty liver disease (NAFLD)

During non-obese NAFLD, abnormal accumulation of lipids and apolipoproteins occurs in liver cells, potentially causing detrimental liver defects. It is hypothesized that misregulation of intracellular trafficking causes lipid accumulation due to impaired secretion of lipo-protein particles containing cholesterol and lipids.

Thus, Peng et al speculated that proteins involved in regulating hepatocyte intracellular trafficking could provide a molecular basis for non-obese NAFLD. To investigate this hypothesis, the authors focused on the Golgi protein GP73. This protein contains a putative Rab GTPase-activating protein (GAP) activity. The physiological role of GAPs is to deactivate Rab-proteins. Since Rabs are master regulators of intracellular vesicular trafficking, their inactivation by GAPs decreases intracellular transport. Peng et al. investigated whether GP73 may cause a reduction in lipid and cholesterol secretion via Rab deactivation.

In their study, the authors identified the Rab-protein Rab23 as a putative target for GP73. Biochemical analysis indeed suggested a GAP-activity of GP73 towards Rab23, leading to its deactivation. Mouse experiments and patient analyses demonstrated that GP73 overexpression indeed diminished cholesterol and lipid secretion with concomitant accumulation of these metabolites in the liver cytosol.

Furthermore, the authors demonstrated that the drug Metformin – used for Type 2 Diabetes treatment – can inhibit GP73 activity in vitro, suggesting that the lipid level reducing effect of this compound may be due to stimulating lipid and cholesterol secretion.

The manuscript is interesting and thought-provoking and could be suited for Nature Communications. The biological and systemic implications of the authors findings will need to be evaluated by experts from the field. I am focusing on the evaluation of the biochemical data here. Nevertheless, I think that a direct proof for GP73 targeting Rab23 in vivo or in cellulo is required for making this paper ready for Nature Communications (see my specific comments below).

Major comments:

- The biochemical results suggest that GP73 could physiologically act via deactivating Rab23. However, this conclusion is only a correlative one and no direct cause for Rab23 deactivation by GP73 in vivo is presented. GP73 may have another Rab-target in vivo. A direct proof of this signalling axis would be required to establish the mechanism of action of GP73 (perhaps by analyzing Rab23-GP73 colocalization and/or by demonstrating the decrease of Rab23:GTP-levels upon GP73-overexpression or increase in Rab23:GTP-levels upon GP73 knock-down). This is

particularly important since TBC-domain GAPs are known to be notoriously unspecific for Rab proteins in vitro (presumably because they share the same catalytic mechanism).

- The findings of the authors suggest that Metformin may act via inhibition of GP73. Even though this may be true, detailed additional studies will be required to prove this mode-of-action also in vivo. At this point, the authors should at least carefully discuss what is known about putative or proven Metformin-targets in the literature. Also, the authors should comment (or experimentally address) whether Metformin-binding is specific to GP73 or whether other (Rab) GAPs could also be inhibited by it. Has Metformin been shown to affect protein-protein-interactions in general and could this be an unspecific effect?

- Fig. 8a+b: How can the authors make sure that the binding of Metformin to GP73 and the inhibition of GP73-GAP-activity by Metformin is a GP73-specific process? Hypothetically, Metformin might impair any protein-protein interaction, not just GP73-interactions. Ideally, a control experiment with another Rab-GAP pair (e.g. Rab1 and TBC1D20) would be included and demonstrate that Metformin does not bind and has no influence on GAP-mediated Rab-deactivation.

Minor comments:

-Figure 1c: Please include a further control, namely GP73 included with GTP (no Rab23). This should establish that the increase in phosphate is not due to a GTP-hydrolytic activity of GP73 or contaminants from the protein preparation.

- Fig. 1c and related results: I could not find the description of calculating k_{cat}/K_m . Please include the equation and procedure in the material and methods section. Furthermore, the quality of non-linear curve fitting of enzymatic time-traces should be presented, at least in the supplement.

- for all biochemical assays (e.g. Fig. 1c, 1d): Please provide concentrations of the proteins, preferably in the figure captions. For example, I could not find the concentration of GP73 in Fig. 1c and therefore I am unable to estimate whether the k_{cat}/K_m has been calculated correctly.

- Please explain in the introduction the rationale for hypothesizing an involvement of GP73 in NAFLD. Why had the authors the idea that GP73 could be involved in regulating Apo and ApoB100 secretion?

- line 138: Please provide reference in which these motifs have been identified first.

- line 139: Please delete "novel".

- line 140-141: Please provide a rationale why the focus has been on these 13 Rab proteins and why other Rabs have not been included in the screen. Furthermore, please mention in the discussion that hypothetically GP73 may act on another so far untested Rab-protein instead of Rab23.

- line 163, line 210: Please correct: GP73 does not have GTP hydrolytic activity (it does not promote GTP hydrolysis directly, i.e. in the absence of Rab23), but has GAP activity.

- lines 333-334: Please mention how the GP73 plasma levels have been determined experimentally (mRNA-levels? Direct protein quantification?).

- lines 382-385: Please provide reference to this claim.

- method section "Recombinant proteins and GAP assay": Please provide detailed buffer and storage conditions for the proteins used. Of particular importance are the GAP-assay conditions (e.g. Mg^{2+} concentration). Please provide this information as detailed as possible.

REVIEWER COMMENTS

Point to point response to reviewer's comments:

Reviewer #1 (Remarks to the Author):

1. Please provide more details on species and experimental set up used in the study in the abstract.

Details on species and experimental set up used in the study were added in the abstract marked in red in the revised version as below:

The prevalence of non-obese nonalcoholic fatty liver disease (NAFLD) is growing worldwide with unclear etiology and pathogenesis. Here, we show GP73, a Golgi protein up-regulated in livers from patients with a variety of liver diseases, exhibited Rab GTPase-activating protein (GAP) activity regulating ApoB export. Chronic elevations of hepatocyte GP73 in mice triggered non-obese NAFLD highly dependent on its GAP activity. The metabolite profiling of GP73-high livers revealed trends toward the accumulation of lipid metabolites contributing to cytotoxicity and inflammation. Common and specific features of liver gene expression signatures associated with prolonged GP73 elevation and high-fat-diet (HFD) were revealed. Metformin inactivated GP73 GAP activity and alleviated non-obese NAFLD induced by GP73. Notably, GP73 was pathologically elevated in non-obese NAFLD individuals, and GP73 blockade improved whole-body metabolism in non-obese NAFLD mouse model. These findings reveal a pathophysiological role of GP73 in triggering non-obese NAFLD and may offer an opportunity for clinical intervention.

2. There are several statements in the introduction that lack references e.g., dietary pattern of lean patients with NAFLD, microbiota in lean NAFLD patients. Indeed, there are by now several studies in mice that show that the dietary composition e.g., the amount and kind of fat as well as sugar might be critical. The same of intestinal barrier dysfunction and LPS shown to also affect lipid export.

References including dietary compositions in the pathogenesis of lean NAFLD patients and intestinal barrier dysfunction in affecting lipid accumulation has been added in the introduction part marked in red in the revised version as below:

Dietary intake, such as fructose sweetened beverages, soft drinks, and increased

dietary cholesterol, may play an important role in the pathogenesis of non-obese NAFLD^{5,6}. Specific gut microbiota compositions have been observed in lean patients with NAFLD⁷.

Intestinal barrier damage, including the destruction of the intestinal epithelium, the growth of bacteria in the small intestine, and an increase in the level of LPS, is associated with hepatocyte lipid accumulation and apoptosis⁸.

5 Kim, D. & Kim, W. R. Nonobese Fatty Liver Disease. *Clin Gastroenterol Hepatol* 15, 474-485, doi:10.1016/j.cgh.2016.08.028 (2017).

6 Zelber-Sagi, S. et al. Long term nutritional intake and the risk for non-alcoholic fatty liver disease (NAFLD): a population based study. *J Hepatol* 47, 711-717, doi:10.1016/j.jhep.2007.06.020 (2007).

7 Lee, G. et al. Distinct signatures of gut microbiome and metabolites associated with significant fibrosis in non-obese NAFLD. *Nat Commun* 11, 4982, doi:10.1038/s41467-020-18754-5 (2020).

8 Miele, L. et al. Increased intestinal permeability and tight junction alterations in nonalcoholic fatty liver disease. *Hepatology* 49, 1877-1887, doi:10.1002/hep.22848 (2009).

3. Metformin is a drug affecting many pathways, starting with intestinal microbiota, intestinal barrier, glucose metabolism in liver, PAI-1 expression... maybe another drug being more specific would be a better choice for a proof of concept. The same for berberine, where molecular mechanisms of action are not yet fully understood.

In the discussion part, pathways and proven Metformin-targets were added and marked in red as below:

Metformin is a first-line drug for the treatment of type 2 diabetes that inhibits hepatic glucose production through AMPK-dependent and AMPK-independent mechanisms. Direct inhibition of mitochondrial glycerol-3-phosphate dehydrogenase and mitochondrial respiratory chain complex 1 have been proposed for the acute inhibition of gluconeogenesis by metformin. Emerging evidence suggests that metformin could contribute to improvements in obesity-associated insulin sensitivity through various resident immune cells in metabolic organs and gut microbiota.

To address whether Metformin-binding is specific to GP73, the binding of metformin with TBC1D20 and the effect of metformin on TBC1D20-mediated Rab1b-deactivation were monitored. To show whether metformin affect protein-protein-interactions in general, the interaction of Rab1b+ TBC1D20 and GP73+Rab23 in the presence or absence of metformin was monitored.

To be more specific for a proof of concept, GP73 blockage by siRNA was used in non-obese NAFLD mice model induced by high fat, high cholesterol, and cholate (HFHCC) diet. In the intervention protocol, knockdown of liver GP73 significantly improved whole-body metabolism. All the results were added to the results part in the revised version as below.

Figure7B

Figure7C

Figure S9A

Figure 9

4. Please provide methods in the main body of the manuscript.

Some parts of the methods were included in the main body in the revised version.

5. Please provide information of power-calculation for both human and mouse studies.

The sample size for mice metabolic study we set is $n > 5$ in each group and power was calculated by PASS version 15.0.5. The majority of the power values for mice studies were over 0.8. For human study in Figure 8C, the power is 0.9708. We can add these information to the materials and methods part if needed.

6. Ultrasound is not a very reliable way to stage NAFLD, the same for a self-reported assessment of alcohol intake etc.

9. Please provide information on visceral adipose tissue content in patients and controls (ratio of adiponectin and leptin, MRIs or something alike).

It is true that ultrasound is not a very reliable way to stage NAFLD, so we tried hard to recruit these non-obese NAFLD subjects in the past two months, and 14 healthy control and 14 non-obese NAFLD patients were enrolled and diagnosed with MRI, informations about MRI hepatic fat fraction (MRI-HFF), level and ratio of adiponectin and leptin, and serum GP73 levels were added to table S1 and Figure 8C-F in the revised version as below:

Table S1. Characteristics of the study participants based on obesity status.

Characteristics	Non-obese controls (n=14)	Non-obese NAFLD (n=14)	P value
Age (years)	39.2±13.6	40.1±10.4	ns
Gender (male, %)	10 (71.4)	9 (64.3)	ns
BMI (kg/m ²)	21.5±3.5	23.5±1.4	ns
WC (cm)	79.5±6.5	81.3±8.4	ns
Glucose (mmol/L)	5.1±0.5	5.4±1.7	<0.05
SBP (mmHg)	116.4±9.0	121.5±54.2	ns
DBP (mmHg)	68.0±6.6	69.6±13.8	ns
WBC (x10 ⁹ /L)	5.3±1.0	6.8±3.3	<0.05
Hb (g/L)	127.9±15.8	144.5±64.0	<0.001
PLT (x10 ⁹ /L)	227.5±41.1	259.6±130.1	ns
TG (mmol/L)	0.7±0.2	1.9±1.6	<0.0001
CHO (mmol/L)	4.9±0.7	4.8±0.9	ns
LDL-C (mmol/L)	2.6±0.6	2.6±0.7	ns
HDL-C (mmol/L)	1.7±0.3	1.2±0.4	<0.0001
ALT (U/L)	16.7±7.3	37.6±28.8	<0.0001
AST (U/L)	21.5±17.8	24.2±9.8	<0.01
UA (μmol/L)	305.1±74.9	426.9±85.3	<0.0001
Adiponectin (μg/mL)	8.6±5.9	5.3±1.3	<0.01

Leptin (ng/mL)	7.7±3.1	8.5±2.9	ns
A/L (×10 ³)	1.3±1.0	0.7±0.5	<0.05
MRI-HFF (%)	3.0±2.4	32.1±14.8	<0.0001

Abbreviations: NAFLD, nonalcoholic fatty liver disease; BMI, body mass index; WC, waist circumference; SBP, systolic blood pressure; DBP, diastolic blood pressure; WBC, white blood cell; Hb, hemoglobin; PLT, platelet; TG, triglyceride; CHO, cholesterol; HDL, high-density lipoprotein; LDL, low-density lipoprotein; ALT, alanine aminotransferase; AST, aspartate aminotransferase; UA, uric acid; A/L, ratio of adiponectin and leptin; MRI-HFF, magnetic resonance imaging hepatic fat fraction.

The data are expressed as the means ± SDs or numbers (percentages).

The statistical analyses (*P* value) were performed by comparing non-obese controls vs. non-obese NAFLD by Spearman chi-square and unpaired Student's *t*-tests.

Figure 8C

Figure 8D

Figure 8E

Figure 8F

7. Was fasting of mice started a 9 o'clock or at 3 o'clock at night?

For measurement of fasting blood glucose, mice were fasted for 6 h and the fasting was started at 9:00 am. This was clarified in the methods part in the revised version marked in red.

8. How specific the overexpression of GP73 for the liver?

10. How specific is the overexpression of GP73 in the mice treated with the adenovirus in hepatocytes? How about other liver cells and tissues?

The specific overexpression of AAV-GP73 for the liver and hepatocytes was

monitored by qRT-PCR and immunoblotting. Three weeks after AAV-GP73 administration, multiple tissues were harvested for evaluating the expression profile of GP73. As we can see, AAV-GP73-induced GP73 expression occurred mainly in the liver, indicating that the tail vein injection essentially limited the adenovirus target location to the liver. In addition, GP73 expression was mainly induced in hepatocytes in livers from AAV-GP73-infected mice. All the results were added to the results part in the revised version as below:

Figure S2

A

B

11. The authors show ApoB100, what about ApoB48?

ApoB48 was analyzed and the result was added to the results part in the revised version as Figure S1F as below:

Figure S1F

12. The description of the animal procedures is rather unclear in the methods section.

Details of the animal procedures was added to the materials and methods part in the revised version as below:

Animals

Male C57BL/6N WT mice were purchased from SPF Biotechnology (Beijing, China). All mice were group-housed conventionally under a 12-h light/dark cycle in an animal facility for 3 days before any procedures. All animal experiments were conducted at the AMMS Animal Center (Beijing, China) and were approved by the Institutional Animal Care and Use Committee.

For AAV-GP73 injections, AAVs encoding mouse GP73, GP73 R248K and the Q310A mutant were constructed and propagated by Hanbio, Inc., Shanghai, China. Mice were injected with 3×10^{11} vg intravenously every six months.

For VLDL secretion, mice were fasted overnight and then intravenously injected with tyloxapol (400 mg/kg body weight). Blood was taken at the indicated time points, and plasma TG levels were determined according to the relevant kit manufacturer's instructions.

For insulin and glucose tolerance tests, mice were fasted for 6 h. Fasted blood glucose was measured in blood collected from the tail vein by tail snipping, and then glucose (1.5 g/kg body weight) and human insulin (0.75 U/kg body weight) were administered intravenously to conscious animals. Glucose was measured in blood taken from the tail vein at 0, 30, 60, 90 and 120 min post injection.

To establish an HFHCC-induced non-obese NAFLD model, male C57BL/6N mice were maintained on a regular chow diet or fed a HFHCC diet for 4 weeks. During this time, GP73 or control siRNA oligos were administered at a dose of 0.2 nmol/g to mice via tail vein injection twice a week. Mice were then euthanized for collection of plasma and liver tissues after 4 weeks of feeding. Samples were snap frozen in liquid nitrogen and stored at -80°C until analysis.

Chemical modifications siRNA for in vivo study

In the sense strand of siRNA, the 3' terminus was modified by cholesterol and 4 thiols, the 5' terminus was modified by 2 thiols, and the whole strand was modified by 2'-O-methyl-modified (2' OMe-modified). All the siRNAs were obtained from GenePharma.

The siRNAs targeting GP73 had the following target sequences:

siGP73 (275): 5'- CCUGGUGGCCUGUGUUAUUTT -3'

Scrambled siRNA oligonucleotides from siGP73 were used as a control (siCtrl).

13. How was food intake of mice with liver-GP73 high? Also, how was body weight in older mice? Did animals continue to gain less weight or lose weight?

Food intake in liver-GP73 high mice was similar between the two groups from month 5 to 6 after AAV injection. During this period, reduced body weights in GP73-liver-high mice were identified. Body weight of liver-GP73 high mice continue to lose weight throughout the experimental duration (12 months). This part was added to the results part in the revised version as Figure S2F as below:

Figure S3E

Figure S3F

14. Why were different ages of mice used? Also, did liver damage progress? Did

animals develop at older age and prolonged overexpression of GP73 develop fibrosis?

We tried to analyze the development of metabolic abnormality as GP73 expression times prolonged. As we can see, liver-GP73-high mice began to exhibit higher fasting blood glucose levels at month 4.5 after AAV injection. Notably, the plasma levels of AST and ALT increased gradually and significantly at months 6 and 12, indicating liver damage progresses throughout the experimental duration (12 months). In addition, body weight continued to lose and mild fibrosis were identified at months 12 after AAV-GP73 injection. Nevertheless, 4.5 months was a starting time point for phenotype assay. To avoid confusion, weeks were replaced by months in the revised version.

15. The comparison to HFD is not adequate as these mice normally become overweight and therefore metabolism may markedly differ. Also, after 12 months HFD fed mice should have beginning fibrosis. An n of 3 is very low. 12 months HFD fed mice began to display fibrosis.

It is true. Besides the comparison of liver gene expression signatures associated with

prolonged GP73 elevations and obese NAFLD, non-obese NALFD should also be monitored. Unfortunately, we do not have enough qualified GP73-high liver samples and it took 12 months to repeat the whole process. We thus lower the tones and rephrase the word from “comparison” to “revealed”: Common and specific features of liver gene expression signatures associated with GP73-induced non-obese NAFLD and diet-induced obese NALFD are revealed. Liver-GP73-high mice displayed fibrosis at month 12 after AAV injection, a metabolic phenotype aligned with those observed in HFD-induced NAFLD. For costly lipidomics and gene expression microarray assay, we used 3 mice in each group, this number was really low, but correlation analysis was monitored.

Figure 6B

16. GP73 expression in human liver, not only plasma should be presented as the origin of GP73 cannot be distinguished this way. Also, it seems that levels vary considerable suggesting that other factors might be more important in these patients.

Since it is impossible to conduct GP73 expression in human livers in non-obese subjects, we conducted non-obese mice model by HFHCC feeding, and found

elevated GP73 expression was restricted to livers in these mice at weeks 4 after feeding. Not only plasma GP73 cannot represent the origin of GP73, but also that elevated GP73 expression in the livers does not always reflect in the plasma. Nevertheless, we do see low serum GP73 levels in non-obese subjects. At present, we could not exclude the possibility that other factors might be more important in these patients. So we lower our tone and changed the title from “mediating to contributing”:

GP73 is a potent TBC-domain GAP contributing to the pathogenesis of non-obese nonalcoholic fatty liver disease.

17. A group of mice only receiving metformin and not treated with AAV-GP73 should be shown.

Data of mice receiving metformin were added to the results part in the revised version as below:

Figure 7

18. As the authors mention the role of microbiota and intestinal barrier in the introduction and the liver and gut have been shown to interact, did the treatment with AAV-GP73 somehow affect intestinal barrier function?

We did HE staining in intestinal from AAV or AAV-GP73 mice at month 5 after injection, and intestinal barrier is normal. We thus proposed that the onset of non-obese NAFLD by hepatocyte GP73 expression is independent of microbiota and intestinal barrier. However, we do not exclude the possible contribution of microbiota and intestinal barrier upon prolonged GP73 elevation. We could add the result below if needed:

Reviewer #2 (Remarks to the Author):

The present study investigates the role of GP73 in the pathogenesis of NAFLD in a mouse model. Data include lipid analysis either by enzymatic testing and lipidomic workflows. Based on my expertise I will specifically comment on lipidomics. Unfortunately, the lipidomic methods are not described in sufficient detail. However, based on the current description, the applied method does not meet current state of the art for lipid species quantification.

The widely targeted lipidomics profiling was performed by Metware Biotechnology (Wuhan, China), a commercialized company focus on lipidomics analysis that many peer-reviewed articles have used (Supplementary References 6: J Agric Food Chem. 2020 Jun 3;68(22):6142-6153. doi: 10.1021/acs.jafc.0c01778. Epub 2020 May 21; Supplementary References 7: Clin Transl Med. 2020 Sep;10(5):e189. doi: 10.1002/ctm2.189; Supplementary References 8: Cell Biosci. 2021 May 22;11(1):95. doi: 10.1186/s13578-021-00604-6). It is our problem that we did not point this out in the methods part in the old version. The detailed methodology of the lipidomics was consistent with previous reported studies. Some informations belongs to commercial confidential, however, we added informations to the lipidomics in the methods part and the supporting table 2 and 3 as detailed as possible.

- Samples were extracted including methanol, MTBE and internal standard mixture. Suitable lipid extraction is key for accurate lipid quantification. Please add details of the procedure including sample and solvent volumes and composition of the internal standard mixture.

Details of the procedure were added to the methods part as below:

Briefly, 50 mg liver samples were homogenized in a 1 mL mixture (methyl tert-butyl ether: methanol =3:1, V/V, mixed solution containing internal standard). The mixture was then vortexed for 1 min and centrifuged for 10 min (4 °C, 12000 rpm). A volume of 500 µL of the supernatant was collected and redissolved in 100 µL of mobile phase B (acetonitrile/isopropanol (10/90, V/V) containing 0.1% formic acid and 10 mmol/L ammonium formate) for LC-MS/MS analysis.

Table S2: Composition of the internal standard mixture:

Internal Standard	CAS	Q1	Q3	RT (min)	ION MODE
PC(13:0/13:0)	71242-28-9	650.5	184.0	5.15	positive
PE(12:0/12:0)	59752-57-7	580.4	439.4	4.74	positive
PG(12:0/12:0)	322647-27-8	611.4	439.4	4.36	positive
PI(16:0/16:0)	34290-57-8	809.5	255.2	5.86	negative
PS(14:0/14:0)	105405-50-3	680.5	257.2	5.22	positive
LPC(12:0)	20559-18-6	527.3	184.0	2.35	positive
LPE(14:0)	123060-40-2	426.3	285.3	1.53	positive
Cer(d18:1/4:0)	74713-58-9	370.3	264.2	3.545	positive
CE(17:0)	24365-37-5	656.6	369.4	14.15	positive
DG(12:0/12:0)	60562-15-4	474.4	257.4	5.365	positive
TG(12:0/12:0/12:0)	538-24-9	656.6	439.5	9.95	positive
FFA(16:0)-d31	39756-30-4	286.3	286.3	4.25	negative

- It is unclear, how data were measured (which mass transitions; were isomeric and isobaric interferences considered; how was quantification performed etc.). Please add these details and suppl. tables including mass transition, RT, internal standard for all measured lipid species.

Details of the methods and tables including mass transition, RT and internal standard were added in the revised version:

Analyst 1.6.3 software (AB Sciex) was used to process the raw mass spectrometry data. The analytical conditions and detailed work parameters were presented as described in reported literatures (Supplementary References 6: J Agric Food Chem. 2020 Jun 3;68(22):6142-6153. doi: 10.1021/acs.jafc.0c01778. Epub 2020 May 21;

Supplementary References 7: Clin Transl Med. 2020 Sep;10(5):e189. doi: 10.1002/ctm2.189; Supplementary References 8: Cell Biosci. 2021 May 22;11(1):95. doi: 10.1186/s13578-021-00604-6):95. doi: 10.1186/s13578-021-00604-6).

Qualitative analysis of the MS and MS/MS mass spectrometric data was performed by comparison of the accurate precursor ions (Q1), product ions (Q3), retention time (RT), and fragmentation patterns with those obtained by injecting standards at the same conditions or the homemade database MWDB (MetWare, China). The characteristic ions of each metabolite were screened using a QQQ mass spectrometer to obtain the signal strengths. Integration and correction of chromatographic peaks was performed using MultiQuant software (AB Sciex). The corresponding relative metabolite contents were represented as chromatographic peak area integrals. Isomers and isobaric can be separated by RT interval or by collecting characteristic ion (Q3). In addition, accurate masses of features representing significant differences were searched against the Kyoto Encyclopedia of Genes and Genomes (KEGG) databases.

Table S3: mass transitions and RT of partial measured lipids:

Compounds	Class	Exact mass (Da)	ION mode	Ionization	Q1 (m/z)	Q3 (m/z)	RT (min)
(±)12-HEPE	Eicosanoid	318.219495	negative	[M-H]-	317.2	179.2	1.39
FFA(22:6)	FFA	328.24023	negative	[M-H]-	327.2	327.2	3.51
LPC(18:0/0:0)	LPC	523.363792	negative	[M+COOH]-	568.4	283.2	3.64
PC(16:1/18:2)	PC	755.546508	negative	[M+COOH]-	800.5	279.2	5.71
PE(20:5/18:1)	PE	763.515208	negative	[M-H]-	762.5	301.2	5.95
PG(16:1/22:6)	PG	792.494139	negative	[M-H]-	791.5	253.2	5.02
PI(18:0/20:4)	PI	886.557134	negative	[M-H]-	885.5	303.2	6.04
Free carnitine	CAR	161.1056065	positive	[M+H]+	162.1	85.1	0.725
Linoleyl-carnitine	CAR	423.3360965	positive	[M+H]+	424.3	85.1	1.96
CE(18:1)	CE	650.6002	positive	[M+NH4]+	668.6	369.4	13.93
CerP(d18:1/18:0)	CerP	645.509727	positive	[M+H]+	646.5	264.2	10.595
Cer(t18:0/24:1)	Cer	665.6322	positive	[M+H]+	666.6	282.2	8.7
DG(14:0/18:2/0:0)	DG	564.475376	positive	[M+NH4]+	582.5	285.5	6.76
LPC(12:0/0:0)	LPC	439.269892	positive	[M+H]+	440.3	184.0	0.785
LPE(P-20:0/0:0)	LPE	493.3532	positive	[M+H]+	494.4	353.4	3.315
LPS(16:0/0:0)	LPS	497.275372	positive	[M+H]+	498.3	313.3	1.835
MG(16:0/0:0/0:0)	MG	330.277011	positive	[M+NH4]+	348.3	257.3	3.905
PC(16:0/16:0)	PC	733.562158	positive	[M+H]+	734.6	184.0	6.7

PS(22:6/16:0)	PS	807.505038	positive	[M+H] ⁺	808.5	623.5	5.52
SM(d18:2/20:1)	SM	754.598876	positive	[M+H] ⁺	755.6	184.0	6.11
TG(14:0/16:0/16:0)	TG	778.705042	positive	[M+NH ₄] ⁺	796.7	523.7	11.9
TG(18:0/20:4/22:6)	TG	954.767642	positive	[M+NH ₄] ⁺	972.8	627.8	11.39

- Was the method including lipid extraction validated? If yes, please include references or data concerning reproducibility, LOD etc.?

Lipid extraction references concerning validation were added to the revised version in the methods part:

Supplementary References 9: Lipid extraction by methyl-tert-butyl ether for high-throughput lipidomics. The Journal of Lipid Research, 2008, 49(5):1137-1146.

Reproducibility of the lipidomics was added to the method part marked in red in the revised version as below:

The calibration and quality control (QC) samples were prepared with the mixed mixing sample extracts prior to sample analysis. Every 10 samples to be analyzed were separated by one QC sample for the duration of the detection to monitor repeatability during the analysis. The high overlaps of the total ion flow current diagram, consistently retention time, and peak strength between different QC samples indicates that the signal stability of the mass spectrum is good at different times.

- Figure S3 shows a heat map of regulated lipid species. Lipid species and molecular lipid species (i.e. acyl chains) were annotated. I doubt that sn-positions are proven by experimental data. Please use the hierarchical concept and annotation of the latest shorthand nomenclature for lipid species (DOI: 10.1194/jlr.S120001025). Please describe how fold changes were calculated (based on counts or nmol?). In part annotations are not correct e.g. PC 42:2 (O-20:1/22:1) should read PC O-42:2.

The latest shorthand nomenclature for lipid species were used and new figure S3 was added to the revised version as below.

- Line 232: "Lipid levels were normalized to total lipid contents and transformed as log₂ fold-change relative to the AAV-V control." This kind of normalization most likely leads to wrong conclusions due to substantially different TG content of the samples. Consequently, subsequent data interpretation is presumably not valid. Comparison

should be based on molar concentrations without normalization to total lipid content (especially when samples show a high degree of lipid storage). Most likely, downregulated PC and PE fraction (Fig. 4E) is derived from normalization – I guess related to wet weight or protein these phospholipid classes remain unchanged.

Sorry for the confusion. Lipid levels were peak areas of corresponding lipid substances from livers of same weight, which was clarified in the methods part marked in red.

Reviewer #3 (Remarks to the Author):

Major comments:

- The biochemical results suggest that GP73 could physiologically act via deactivating Rab23. However, this conclusion is only a correlative one and no direct cause for Rab23 deactivation by GP73 in vivo is presented. GP73 may have another Rab-target in vivo. A direct proof of this signalling axis would be required to establish the mechanism of action of GP73 (perhaps by analyzing Rab23-GP73 colocalization and/or by demonstrating the decrease of Rab23:GTP-levels upon GP73-overexpression or increase in Rab23:GTP-levels upon GP73 knock-down). This is particularly important since TBC-domain GAPs are known to be notoriously unspecific for Rab proteins in vitro (presumably because they share the same catalytic mechanism).

Sorry we did not demonstrate the Rab23:GTP levels in GP73 overexpression cells, we tried several ways to assess intracellular Rab23-GTP levels, but failed. As a complement, the interaction between Rab23 and GP73 by immunoprecipitation in the presence or absence of metformin was conducted along with Rab1b and TBC1D20 controls. In addition, although GP73 showed the highest GAP activity toward Rab23, and the activity of Rab23 to promote the secretion of ApoB was in consistent with the phenotype that GP73 overexpression led to ApoB secretion impairment, some other cargos, including albumin, was stimulated both by Rab23 and GP73. Therefore, other Rabs (tested and untested) targeted by GP73 may also be responsible for the observed differences. Thus, our findings focus on the contribution of GP73 GAP activity on non-obese NAFLD, not via deactivating Rab23 as its only substrate. We clarified this point in the discussion part and added new figures 7B-C and S9A in the revised version.

- The findings of the authors suggest that Metformin may act via inhibition of GP73. Even though this may be true, detailed additional studies will be required to prove this mode-of-action also in vivo. At this point, the authors should at least carefully discuss what is known about putative or proven Metformin-targets in the literature. Also, the authors should comment (or experimentally address) whether

Metformin-binding is specific to GP73 or whether other (Rab) GAPs could also be inhibited by it. Has Metformin been shown to affect protein-protein-interactions in general and could this be an unspecific effect?

- Fig. 8a+b: How can the authors make sure that the binding of Metformin to GP73 and the inhibition of GP73-GAP-activity by Metformin is a GP73-specific process? Hypothetically, Metformin might impair any protein-protein interaction, not just GP73-interactions. Ideally, a control experiment with another Rab-GAP pair (e.g. Rab1b and TBC1D20) would be included and demonstrate that Metformin does not bind and has no influence on GAP-mediated Rab-deactivation.

In the discussion part, pathways and proven Metformin-targets were added and marked in red as below:

Metformin is a first-line drug for the treatment of type 2 diabetes that inhibits hepatic glucose production through AMPK-dependent and AMPK-independent mechanisms. Direct inhibition of mitochondrial glycerol-3-phosphate dehydrogenase and mitochondrial respiratory chain complex 1 have been proposed for the acute inhibition of gluconeogenesis by metformin. Emerging evidence suggests that metformin could contribute to improvements in obesity-associated insulin sensitivity through various resident immune cells in metabolic organs and gut microbiota.

To address whether Metformin-binding is specific to GP73, the binding of metformin with TBC1D20 and the effect of metformin on TBC1D20-mediated Rab1b-deactivation were monitored. To show whether metformin affect protein-protein-interactions in general, the interaction of Rab1b+ TBC1D20 and GP73+Rab23 in the presence or absence of metformin was monitored.

To be more specific for a proof of concept, GP73 blockage by siRNA was used in non-obese NAFLD mice model induced by high fat, high cholesterol, and cholate (HFHCC) diet. In the intervention protocol, knockdown of liver GP73 significantly improved whole-body metabolism. All the results were added to the results part in the revised version as below.

Figure 7B

Figure 7C

Figure S9A

Figure 9

Minor comments:

-Figure 1c: Please include a further control, namely GP73 included with GTP (no Rab23). This should establish that the increase in phosphate is not due to a GTP-hydrolytic activity of GP73 or contaminants from the protein preparation.

GP73 included with GTP (no Rab23) was conducted and the result was added to the revised version as Figure 1C.

- Fig. 1c and related results: I could not find the description of calculating k_{cat}/K_m . Please include the equation and procedure in the material and methods section. Furthermore, the quality of non-linear curve fitting of enzymatic time-traces should be presented, at least in the supplement.

The description of calculating K_{cat}/K_m and the non-linear curve fitting of enzymatic time-traces were added in the material and methods section as below:

GAP assay details

A GAP assay using an EnzChek Phosphate Assay Kit (Invitrogen, E12020) and kinetics determinations were performed in strict accordance with a previously described procedure (Nature. 2006 Jul 20;442(7100):303-6. doi: 10.1038/nature04847.). Briefly, Rabs were loaded with GTP (ThermoFisher Scientific, R0461) by incubating GP73 with a 50-fold molar of GTP at 25 °C for 1 h in 20mM HEPES pH7.5, 150mM NaCl, 5mM EDTA, 1mM dithiothreitol. Free GTP was removed with a desalting column (ThermoFisher Scientific, 89891) pre-equilibrated with 20mM HEPES pH 7.5, 150mM NaCl. The single-turnover kinetics of intrinsic and GAP-accelerated GTP hydrolysis were measured by a continuous enzyme assay for the release of inorganic phosphate with the use of reagents from the EnzChek Phosphate Assay Kit (Invitrogen, E12020). GTP-loaded Rabs were mixed with solutions containing the assay reagents and GAPs. The final solutions contained 20mM HEPES pH 7.5, 150mM NaCl, 0.15mM 2-amino-6-mercapto-7-methylpurine ribonucleoside, 0.75U/ml purine nucleoside phosphorylase, 10mM $MgCl_2$, 20 nM GP73 protein and various concentrations of GTP loaded Rabs. The absorbance at 360nm was monitored with microplate

spectrometer (Tecan, M1000). Data were analyzed by fitting them simultaneously to the pseudo-first-order Michaelis-Menten model function:

$$A(t) = (A_{\infty} - A_0)(1 - e^{-kt}) + A_0$$

$$k_{\text{obs}} = k_{\text{intr}} + \frac{k_{\text{cat}}}{K_M} [GAP]$$

For calculating the catalytic efficiency (k_{cat}/K_M), the observed kinetic (k_{obs}) and the intrinsic rate constant (k_{intr}) were measured by fitting the data into a linear regression model according to the transformation form of the pseudo-first-order Michaelis-Menten model function. The calculation is shown below:

$$A(t) = (A_{\infty} - A_0)(1 - e^{-kt}) + A_0$$

$$A(t) = (A_{\infty} - A_0) - (A_{\infty} - A_0)e^{-kt} + A_0$$

$$A(t) = A_{\infty} - (A_{\infty} - A_0)e^{-kt}$$

$$A(t) - A_{\infty} = -(A_{\infty} - A_0)e^{-kt}$$

$$A_{\infty} - A(t) = (A_{\infty} - A_0)e^{-kt}$$

$$\ln[A_{\infty} - A(t)] = \ln[(A_{\infty} - A_0)e^{-kt}]$$

$$\ln[A_{\infty} - A(t)] = \ln(A_{\infty} - A_0) + \ln(e^{-kt})$$

$$\ln[A_{\infty} - A(t)] = \ln(A_{\infty} - A_0) - kt$$

From this equation, $\ln[A_{\infty} - A(t)]$ was regarded as the response variable and was regressed on the explanatory variable time t . The resulting regression coefficients were the desired rate constants with minus signs in the front. The observed kinetic (k_{obs}) and the intrinsic rate constant (k_{intr}) were then acquired by removing the minus signs, and the value of k_{obs} and k_{intr} was plugging back into the below equation to obtain the catalytic efficiency (k_{cat}/K_M) with the concentration of GTPase-activating protein (GAP) set to be 20 nM:

$$k_{\text{obs}} = k_{\text{intr}} + \frac{k_{\text{cat}}}{K_M} [GAP]$$

The catalytic efficiency (k_{cat}/K_M) and intrinsic rate constant for GTP hydrolysis (k_{intr}) were treated as global parameters.

The non-linear curve fitting of enzymatic time-traces for Figure 1C-E, Figure 7D-E and Figure S9A were presented:

Figure 1

Figure 7D

Figure 7E

Figure S9A

- for all biochemical assays (e.g. Fig. 1c, 1d): Please provide concentrations of the proteins, preferably in the figure captions. For example, I could not find the concentration of GP73 in Fig. 1c and therefore I am unable to estimate whether the k_{cat}/K_m has been calculated correctly.

The concentration of GP73 was 20 nM and this information was added to the figures.

- Please explain in the introduction the rationale for hypothesizing an involvement of GP73 in NAFLD. Why had the authors the idea that GP73 could be involved in regulating Apo and ApoB100 secretion?

The rationale of hypothesizing an involvement of GP73 in NAFLD was added as below:

As a type II transmembrane glycoprotein located in the Golgi, the expression of GP73 in hepatocytes is very limited or undetectable in healthy livers. However, in patients with acute or chronic liver diseases, the expression of GP73 is significantly up-regulated in hepatocytes. We then wanted to assess the metabolic consequences of hepatic overexpression of GP73. The idea that GP73 could be involved in regulating Apo and ApoB100 secretion was added to the results part as below:

Given that Rab23 regulates cargo transport from the ER to the Golgi and the secretion of lipoprotein, we hypothesized that Rab23 deactivation by GP73 might impact hepatocyte lipoprotein secretion.

- line 138: Please provide reference in which these motifs have been identified first.

References were added in the revised version:

20 Pan, X., Eathiraj, S., Munson, M. & Lambright, D. G. TBC-domain GAPs for Rab GTPases accelerate GTP hydrolysis by a dual-finger mechanism. *Nature* 442, 303-306, doi:10.1038/nature04847 (2006).

21 Richardson, P. M. & Zon, L. I. Molecular cloning of a cDNA with a novel domain present in the *tre-2* oncogene and the yeast cell cycle regulators BUB2 and *cdc16*. *Oncogene* 11, 1139-1148 (1995).

- line 139: Please delete “novel”.

“Novel” was deleted.

- line 140-141: Please provide a rational why the focus has been on these 13 Rab proteins and why other Rabs have not been included in the screen. Furthermore, please mention in the discussion that hypothetically GP73 may act on another so far untested Rab-protein instead of Rab23.

A rational why the focus has been on these 13 Rab proteins and why other Rabs have not been included in the screen was added to the results part as below:

To test this hypothesis, we investigated the ability of a recombinant GP73 protein to accelerate GTP hydrolysis. Selected Rab GTPases were mainly located at the plasma and intracellular membranes.

GP73 may act on another so far untested Rab-protein instead of Rab23 was added to the discussion part as below:

Among the 13 assayed mammalian Rab GTPases, GP73 showed the highest GAP activity toward Rab23. In support of our findings, the activity of Rab23 has been reported to promote the secretion of ApoB. However, secretion of cargoes, including albumin, was stimulated both by GP73 and Rab23. Therefore, other Rabs (tested and untested) targeted by GP73 may also be responsible for the observed differences.

- line 163, line 210: Please correct: GP73 does not have GTP hydrolytic activity (it

does not promote GTP hydrolysis directly, i.e. in the absence of Rab23), but has GAP activity.

"GTP hydrolytic activity" was replaced with "GAP activity" and marked in red in the revised version.

- lines 333-334: Please mention how the GP73 plasma levels have been determined experimentally (mRNA-levels? Direct protein quantification?).

Plasma GP73 levels were measured using ELISA kits obtained from Hotgen (No.03.03.0201, Hotgen Biotech Co., Ltd. Beijing, China). This was added to the methods part marked in red in the revised version.

- lines 382-385: Please provide reference to this claim.

The references to the claim were added:

33 Kim, H. J., Lv, D., Zhang, Y., Peng, T. & Ma, X. Golgi phosphoprotein 2 in physiology and in diseases. *Cell Biosci* 2, 31, doi:10.1186/2045-3701-2-31 (2012).

34 Xia, Y. et al. Golgi protein 73 and its diagnostic value in liver diseases. *Cell Prolif* 52, e12538, doi:10.1111/cpr.12538 (2019).

35 Wang, L. et al. Serum Golgi Protein 73 as a Potential Biomarker for Hepatic Necroinflammation in Population with Nonalcoholic Steatohepatitis. *Dis Markers* 2020, 6036904, doi:10.1155/2020/6036904 (2020).

36 Zheng, K. I. et al. Combined and sequential non-invasive approach to diagnosing non-alcoholic steatohepatitis in patients with non-alcoholic fatty liver disease and persistently normal alanine aminotransferase levels. *BMJ Open Diabetes Res Care* 8, doi:10.1136/bmjdr-2020-001174 (2020).

- method section "Recombinant proteins and GAP assay": Please provide detailed buffer and storage conditions for the proteins used. Of particular importance are the GAP-assay conditions (e.g. Mg²⁺ concentration). Please provide this information as detailed as possible.

The buffer of GAP-assay was added in the Materials and methods section:

The final solutions contained 20mM HEPES pH 7.5, 150mM NaCl, 0.15mM 2-amino-6-mercapto-7-methylpurine ribonucleoside, 0.75U/ml purine nucleoside phosphorylase, 10mM MgCl₂, 20 nM GP73 protein and various concentrations of

GTP loaded Rabs. Recombinant proteins were stored at -80 °C.

GAP assay details

A GAP assay using an EnzChek Phosphate Assay Kit (Invitrogen, E12020) and kinetics determinations were performed in strict accordance with a previously described procedure (Nature. 2006 Jul 20;442(7100):303-6. doi: 10.1038/nature04847.). Briefly, Rabs were loaded with GTP (ThermoFisher Scientific, R0461) by incubating GP73 with a 50-fold molar of GTP at 25 °C for 1 h in 20mM HEPES pH7.5, 150mM NaCl, 5mM EDTA, 1mM dithiothreitol. Free GTP was removed with a desalting column (ThermoFisher Scientific, 89891) pre-equilibrated with 20mM HEPES pH 7.5, 150mM NaCl. The single-turnover kinetics of intrinsic and GAP-accelerated GTP hydrolysis were measured by a continuous enzyme assay for the release of inorganic phosphate with the use of reagents from the EnzChek Phosphate Assay Kit (Invitrogen, E12020). GTP-loaded Rabs were mixed with solutions containing the assay reagents and GAPs. The final solutions contained 20mM HEPES pH 7.5, 150mM NaCl, 0.15mM 2-amino-6-mercapto-7-methylpurine ribonucleoside, 0.75U/ml purine nucleoside phosphorylase, 10mM MgCl₂, 20 nM GP73 protein and various concentrations of GTP loaded Rabs. The absorbance at 360nm was monitored with microplate spectrometer (Tecan, M1000). Data were analysed by fitting them simultaneously to the pseudo-first-order Michaelis-Menten model function:

$$A(t) = (A_{\infty} - A_0)(1 - e^{-kt}) + A_0$$

$$k_{\text{obs}} = k_{\text{intr}} + \frac{k_{\text{cat}}}{K_{\text{M}}} [\text{GAP}]$$

For calculating the catalytic efficiency ($k_{\text{cat}}/K_{\text{M}}$), the observed kinetic (k_{obs}) and the intrinsic rate constant (k_{intr}) were measured by fitting the data into a linear regression model according to the transformation form of the pseudo-first-order Michaelis-Menten model function. The calculation is shown below:

$$A(t) = (A_{\infty} - A_0)(1 - e^{-kt}) + A_0$$

$$A(t) = (A_{\infty} - A_0) - (A_{\infty} - A_0)e^{-kt} + A_0$$

$$A(t) = A_{\infty} - (A_{\infty} - A_0)e^{-kt}$$

$$A(t) - A_{\infty} = -(A_{\infty} - A_0)e^{-kt}$$

$$A_{\infty} - A(t) = (A_{\infty} - A_0)e^{-kt}$$

$$\ln[A_{\infty} - A(t)] = \ln[(A_{\infty} - A_0)e^{-kt}]$$

$$\ln[A_{\infty} - A(t)] = \ln(A_{\infty} - A_0) + \ln(e^{-kt})$$

$$\ln[A_{\infty} - A(t)] = \ln(A_{\infty} - A_0) - kt$$

From this equation, $\ln[A_{\infty} - A(t)]$ was regarded as the response variable and was regressed on the explanatory variable time t . The resulting regression coefficients were the desired rate constants with minus signs in the front. The observed kinetic (k_{obs}) and the intrinsic rate constant (k_{intr}) were then acquired by removing the minus signs, and the value of k_{obs} and k_{intr} was plugging back into the below equation to obtain the catalytic efficiency ($k_{\text{cat}}/K_{\text{m}}$) with the concentration of GTPase-activating protein (GAP) set to be 20 nM:

$$k_{\text{obs}} = k_{\text{intr}} + \frac{k_{\text{cat}}}{K_{\text{M}}}[GAP]$$

The catalytic efficiency ($k_{\text{cat}}/K_{\text{m}}$) and intrinsic rate constant for GTP hydrolysis (k_{intr}) were treated as global parameters.

Reviewers' Comments:

Reviewer #1:

Remarks to the Author:

The authors addressed some of my concerns; however, there are some further issues:

While going into detail regarding different effects of metformin similar issues were raised before for berberine. Berberine is an alkaloid extracted from plants and is sold as a plant supplement over the counter with several proposed effects. However, to the knowledge of the reviewer, berberine is not an approved drug for the treatment of diseases as for most diseases the evidence is possible or even insufficient. So, this should be acknowledged in the present manuscript, too.

Please check reference 44, this is not supporting the text.

The authors provide some insight in the power calculations and samples size calculation performed. $N > 5$ seems unusual. One should get a definite sample size number when doing a power calculation. Also, please provide the sample size calculation for the human study. What was the sample size calculation based on?

The use of language needs revision!

Reviewer #2:

Remarks to the Author:

The revised version provides some additional details on lipidomics methods. However, there are still many open questions:

- Lipid extraction: How were the samples homogenized? Aqueous volume? The extraction protocol does not match the referenced Matyash MTBE protocol! Therefore, this procedure should be evaluated at least regarding lipid recovery. Concentrations of the internal standards are missing.
- Table S3 provides some mass transitions. However, a comprehensive overview of all analyzed lipid species is still missing – please add a supplementary Excel sheet. Why was the negative ion mode used for some PC/LPC species and positive ion mode for other PC/LPC species? Annotation of species is inaccurate (and does not represent the latest version of the shorthand) because mass spectrometric data e.g., 755.6 > 184.0 for SM(d18:2/20:1) does only confirm SM 38:3;O2 at species level but not at molecular species level. Moreover, I doubt that sn-positions e.g. for PC 16:1/18:2 are justified by analytical data. Was a retention time model applied for identification?
- The data are not quantitative but based on peak area. Moreover, I could not find a clear description, how data were processed and I am confused, why the authors now removed the previously mentioned normalization? What does “The corresponding relative metabolite contents were represented as chromatographic peak area integrals” mean? How was the internal standard applied? Concerning the methodology, it is not sufficient to refer to several previous studies– such important details should be described comprehensively in the supplementary methods.
- In summary, the lipidomics data are not state of the art quantitative analysis. Therefore, I recommend either to leave out lipidomic data due to their low quality. In addition, far-fetching discussion of the lipidomic results in Figure S4 should be omitted. Alternatively, I recommend performing quantitative comprehensive lipidomic analysis to provide a solid basis for further data analysis like pathway matching as presented in Figure 4.

Reviewer #3:

Remarks to the Author:

The authors present a thoroughly revised version of their manuscript. The biochemical data regarding Rab23 and GP73 appear to be solid and therefore I have no objections on the acceptance of this paper in this regard. However, I would like to emphasize that the biological and cellular data – in particular with respect to lipidomics and the actions of Metformin – will need to be evaluated by experts from the respective fields.

Minor comments

- Line 410, author statement „Therefore, other Rabs targeted by GP73 may also be responsible for the differences.“: The authors have included this statement in response to my first comment in

which I raised caution regarding the specificity of GP73. I am satisfied with their reply, but would recommend that the authors include their full rebuttal statement into the discussion to make absolutely clear that the actual CELLULAR target(s) of GP73 has/have actually not been validated. I believe that this is important in order to prevent misleading other researchers that are interested in following up these studies.

- Line 133: Author statement „Selected Rab GTPases were mainly located at the plasma and intracellular membranes.” Please provide an explanation/ justification as to why the selected Rab GTPases were restricted to this particular localization.

- the authors must check the manuscript and in particular the figures for correct unit definitions. E.g., kcat, not Kcat; kintr, not Kintr; s-1, not S-1.

- Figure 1D: The value of kcat/KM is incorrect (see also text): It should be “10³” not “10⁻³”. I suggest making it consistent with the value in the main text to avoid confusion (i.e. 2.51×10⁵ M⁻¹ s⁻¹).

Reviewer #1 (Remarks to the Author):

The authors addressed some of my concerns; however, there are some further issues:

While going into detail regarding different effects of metformin similar issues were raised before for berberine. Berberine is an alkaloid extracted from plants and is sold as a plant supplement over the counter with several proposed effects. However, to the knowledge of the reviewer, berberine is not an approved drug for the treatment of diseases as for most diseases the evidence is possible or even insufficient. So, this should be acknowledged in the present manuscript, too.

We deleted the berberine data in the revised version, as using an unapproved drug as a control is not suitable.

Please check reference 44, this is not supporting the text.

Reference 44 is a review article, we rephrased the sentence to: Metformin could contribute to improvements in obesity-associated insulin sensitivity.

The authors provide some insight in the power calculations and samples size calculation performed. $N > 5$ seems unusual. One should get a definite sample size number when doing a power calculation. Also, please provide the sample size calculation for the human study. What was the sample size calculation based on?

In mice experiment, no statistical methods were used to predetermine sample size. We used $n=6$ in each group based on prior publications using comparable methods. For this scenario, the majority of the power values for mice studies were ≥ 0.8 . For the human study, the mean and standard deviation are determined by our preliminary experiment measuring the plasma GP73 levels of 105 healthy peoples and 34 non-obese NAFLD diagnosed by ultrasound. Differences in plasma GP73 levels between healthy peoples and non-obese NAFLD are $\geq 150\%$ with SEM within groups of $\leq 20\%$. Hence, setting the significance level α to be 0.05 and the effect size ≥ 1.4 , the minimum sample size required for a Power $(1-\beta)$ of 0.8 was calculated to be ≥ 10 subjects per group. By recruiting 14 healthy controls and 14 non-obese NAFLD group diagnosed by MRI in the present study, the power value is 0.9991. The power analysis was added to the material and methods part as below:

In mice experiment, no statistical methods were used to predetermine sample size. We used $n=6$ in each group based on prior publications using comparable methods⁴³⁻⁴⁵. For this scenario, the

majority of the power values for mice studies were over 0.8. For the human study, based on our preliminary data, differences in plasma GP73 levels between healthy peoples and non-obese NAFLD are $\geq 150\%$ with SEM within groups of $\leq 20\%$. Hence, setting the significance level α to be 0.05 and the effect size ≥ 1.4 , the minimum sample size required for a Power $(1-\beta)$ of 0.8 was calculated to be ≥ 10 subjects per group.

The use of language needs revision!

The present version has been sent out for editing.

Reviewer #2 (Remarks to the Author):

The revised version provides some additional details on lipidomics methods. However, there are still many open questions:

- Lipid extraction: How were the samples homogenized? Aqueous volume? The extraction protocol does not match the referenced Matyash MTBE protocol! Therefore, this procedure should be evaluated at least regarding lipid recovery. Concentrations of the internal standards are missing.
- Table S3 provides some mass transitions. However, a comprehensive overview of all analyzed lipid species is still missing – please add a supplementary Excel sheet. Why was the negative ion mode used for some PC/LPC species and positive ion mode for other PC/LPC species? Annotation of species is inaccurate (and does not represent the latest version of the shorthand) because mass spectrometric data e.g., $755.6 > 184.0$ for SM(d18:2/20:1) does only confirm SM 38:3;O2 at species level but not at molecular species level. Moreover, I doubt that sn-positions e.g. for PC 16:1/18:2 are justified by analytical data. Was a retention time model applied for identification?
- The data are not quantitative but based on peak area. Moreover, I could not find a clear description, how data were processed and I am confused, why the authors now removed the previously mentioned normalization? What does “The corresponding relative metabolite contents were represented as chromatographic peak area integrals” mean? How was the internal standard applied? Concerning the methodology, it is not sufficient to refer to several previous studies– such important details should be described comprehensively in the supplementary methods.

- In summary, the lipidomics data are not state of the art quantitative analysis. Therefore, I recommend either to leave out lipidomic data due to their low quality. In addition, far-fetching discussion of the lipidomic results in Figure S4 should be omitted. Alternatively, I recommend performing quantitative comprehensive lipidomic analysis to provide a solid basis for further data analysis like pathway matching as presented in Figure 4.

We did not perform quantitative comprehensive lipidomic analysis partly because we do not have available samples on our hand. But most importantly, we do not aim to decipher unique or specific lipid class targeting by GP73, but rather to provide evidence showing that chronic elevations in hepatocyte GP73 trigger lipid accumulation. These have been supported in sufficient detail. We thus deleted the low quality lipidomic data in the revised version.

Reviewer #3 (Remarks to the Author):

The authors present a thoroughly revised version of their manuscript. The biochemical data regarding Rab23 and GP73 appear to be solid and therefore I have no objections on the acceptance of this paper in this regard. However, I would like to emphasize that the biological and cellular data – in particular with respect to lipidomics and the actions of Metformin – will need to be evaluated by experts from the respective fields.

Minor comments

- Line 410, author statement “Therefore, other Rabs targeted by GP73 may also be responsible for the differences.”: The authors have included this statement in response to my first comment in which I raised caution regarding the specificity of GP73. I am satisfied with their reply, but would recommend that the authors include their full rebuttal statement into the discussion to make absolutely clear that the actual CELLULAR target(s) of GP73 has/have actually not been validated. I believe that this is important in order to prevent misleading other researchers that are interested in following up these studies.

The full rebuttal statement was added into the discussion part marked in red in the revised version to strengthen the point that actual cellular target(s) of GP73 has/have actually not been validated.

- Line 133: Author statement "Selected Rab GTPases were mainly located at the plasma and intracellular membranes." Please provide an explanation/ justification as to why the selected Rab GTPases were restricted to this particular localization.

As GP73 is a Golgi-resident protein and traffics to the cell surface, we selected Rab GTPases mainly located at the plasma and intracellular membranes. This sentence along with references was added to the results part marked in red in the revised version.

- the authors must check the manuscript and in particular the figures for correct unit definitions. E.g., kcat, not Kcat; kintr, not Kintr; s⁻¹, not S⁻¹.

Unit definitions have been checked and corrected in the revised version.

- Figure 1D: The value of kcat/KM is incorrect (see also text): It should be "10³" not "10⁻³". I suggest making it consistent with the value in the main text to avoid confusion (i.e. 2.51×10⁵ M⁻¹ s⁻¹).

The value of kcat/KM was corrected in the revised version.

Reviewers' Comments:

Reviewer #1:

Remarks to the Author:

I have not further comments. Very interesting study!

Reviewer #3:

Remarks to the Author:

I have no objections to accepting the paper.